# LANGUAGE MODELS CAN HELP TO LEARN HIGH-PERFORMING COST FUNCTIONS FOR RECOURSE

## ABSTRACT

Algorithmic recourse is a specialised variant of counterfactual explanation, concerned with offering actionable recommendations to individuals who have received adverse outcomes from automated systems. Most recourse algorithms assume access to a cost function, which quantifies the effort involved in following recommendations. Such functions are useful for filtering down recourse options to those which are most actionable. In this study, we explore the use of large language models (LLMs) to help label data for training recourse cost functions, while preserving important factors such as transparency, fairness, and performance. We find that LLMs do generally align with human judgements of cost and can label data for the training of effective cost functions, moreover they can be fine-tuned with simple prompt engineering to maximise performance and improve current recourse algorithms in practice. Previously, recourse cost definitions have mainly relied on heuristics and missed the complexities of feature dependencies and fairness attributes, which has drastically limited their usefulness. Our results show that it is possible to train a high-performing, interpretable cost function by consulting an LLM via careful prompt engineering. Furthermore, these cost functions can be customised to add or remove biases as befitting the domain and problem. Overall, this study suggests a simple, accessible method for accurately quantifying notions of cost, effort, or distance between data points that correlate with human intuition, with possible applications throughout the explainable AI field.

## 1 INTRODUCTION

Algorithmic recourse has emerged as one of the most impactful areas of explainable AI (Karimi et al., 2022). The field focuses on generating actionable counterfactual recommendations to users who were treated unfavorably by automated systems, with the canonical example being a rejected bank loan application, and what actions could be taken by a user to have it accepted in future (Ustun et al., 2019). In such a scenario, a cost function is needed to quantify how much effort a recourse recommendation would take, so that algorithms can consider this during optimisation. Separately, it is worth noting that the field has branched out to consider positive outcomes with gain functions and semifactual recourse (Kenny & Huang, 2024). In either case, these functions must align with human domain knowledge and intuition, so they can inform appropriate recourse selection. In this paper, we focus on cost functions, and show how large language models (LLMs) can be used to largely automate their design while maintaining desirable aspects such as transparency and fairness.

Typically in recourse, a cost function is assumed a priori, often as some variant of an $L_p$ norm on the feature space (Keane et al., 2021). For example, an $L_0$ norm assigns higher cost to recourse recommendations that change more features, although this ignores other factors such as how much they are changed. A (weighted) $L_1$ or $L_2$ norm can incorporate magnitude information, but not pairwise or higher-order interactions between features. These can be added in an ad hoc manner, but are challenging to formalise and combine. As an alternative, we examine if the issue of recourse cost can be addressed in a flexible and scalable way by tapping into the tacit domain knowledge of LLMs. We show how, with the right prompting, LLMs can be consulted to compare the costs of pairs of recourses, creating a labelled dataset for training either neural network cost functions or transparent tree-based ones (Kanamori et al., 2022; Bewley & Lecue, 2022). Our results suggest that future research into cost functions may benefit from the use of LLMs.

## 2 COST FUNCTION DESIDERATA

We begin by considering what constitutes a high-performing cost function for recourse applications. Ideally, a cost function should satisfy many intricate criteria which basic $L_p$ norms cannot, such as variable feature weighting and dependencies. Here, we outline our desiderata grounded in prior literature, which will form the basis for subsequent evaluation.

1. *Feature Cost.* A cost function should have different weighting considerations for each feature in the data. For example, adding an additional credit card is generally easier than increasing your down payment (Rawal & Lakkaraju, 2020).

2. *Relative Cost.* A cost function should weigh the cost of a given change differently at different points in the distribution, if appropriate. For example, going from the 55-60th percentile in an exam score may be easier than going from the 90-95th (Ustun et al., 2019).

3. *Dependent Cost.* It must be possible to represent relevant dependencies between two or more features. For example, applying for college funding is usually easier if you are native to a country rather than an immigrant (Karimi et al., 2022).

4. *Fair Cost.* Cost functions should take into account any fairness properties relevant to a given domain and application (Von Kügelgen et al., 2022). In this paper, we define fairness as the cost function not varying its output if demographic information is mutated.

These desiderata have been extensively discussed in the literature cited above. We do not claim this to be an exhaustive list, but a reasonable starting point.[1]

## 3 METHOD

This section outlines our four-step framework for learning cost functions. First, synthetic recourse examples are generated by randomly perturbing a set of data points subject to actionability constraints. Second, pairs of recourse examples are selected at random for comparison. Third, an LLM is queried to provide ratings (i.e. labels) for these comparisons. Finally, the resultant dataset is used to train a cost function. In this process, we assume access to a capable chatbot LLM which may be queried at liberty, and that the data domain is tabular in nature.

### 3.1 GENERATING SYNTHETIC RECOURSES

Let $\mathcal{D} = \{x_i\}_{i=1}^N \subset \mathbb{R}^d$ denote a given dataset, where each $x_i$ represents a $d$-dimensional feature vector. For our purposes, we benefit from $\mathcal{D}$ being as diverse as possible. We define a stochastic perturbation function $\phi : \mathbb{R}^d \times \mathcal{A} \to \Delta(\mathbb{R}^d)$, where $\mathcal{A}$ denotes a set of actionability constraints (see Appendix A for details). The number of features to be perturbed is problem-specific and will determine the cost function's capabilities in deployment. Here, we randomly select this number from a truncated geometric distribution, which favors perturbations of one feature to focus on sparsity, which is desired in recourse (Keane et al., 2021; Karimi et al., 2022). See Appendix B for details.

For each data point $x_i$ and perturbed feature $f \in \{1, \ldots, d\}$, we apply the following perturbation:

$$x_i'[f] = \begin{cases} \sim \text{Uniform}(\text{categories}_f) & \text{if } f \text{ is categorical} \\ x_i[f] + \epsilon : \epsilon \sim \mathcal{E}_f & \text{if } f \text{ is continuous,} \end{cases} \tag{1}$$

where $\text{Uniform}(\text{categories}_f)$ is a uniform distribution over categories and $\mathcal{E}_f$ is a finite set of perturbations for a continuous feature (positive/negative multiples of the standard deviation across $\mathcal{D}$).

This process generates a set of recourse examples $\mathcal{R} = \{(x_i, x_i')\}_{i=1}^N$, where each $x_i$ represents an original instance and $x_i'$ is the corresponding synthetic perturbation. We use a finite set of perturbation magnitudes for numerical features because it allows a direct comparison between exactly the same change at different parts of a feature distribution. This helps to learn relative differences in

---

[1]Another possible desideratum is *Individual Cost*, whereby even if two individuals have identical feature values, they could still have different ideal recourse recommendations based on their preferences (Nauta et al., 2023; Rawal & Lakkaraju, 2020). However, this is largely a separate human computer interaction (HCI) question, and we are instead focused on the training of the cost function itself.

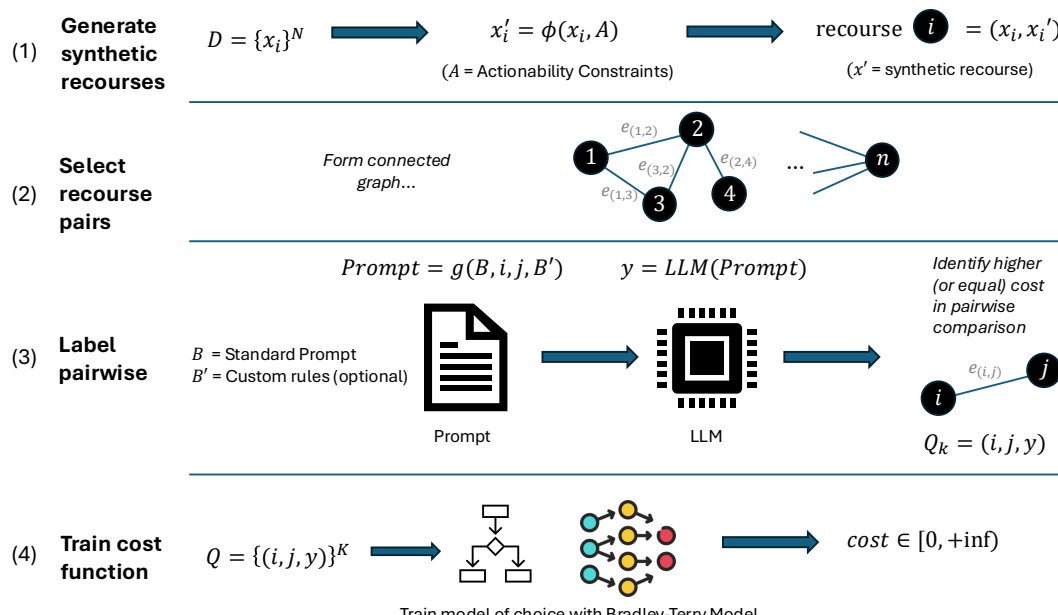

Figure 1: Method Schematic: (1) Each instance in a given dataset is perturbed within actionability constraints to simulate a recourse situation. (2) Pairs of these recourses are selected for comparison in such a way as to from a connected graph (where a path exists between all pairs of recourses). (3) Each edge in the graph is then labeled with an LLM which judges which of the two corresponding recourses takes a higher cost to achieve (or optionally an additional "equal cost" option if specified). (4) The dataset of comparisons is used to train the cost function, in our case either a transparent tree model or an MLP.

cost for the same change, thereby addressing the relative cost criterion (i.e., Desideratum 2). The parameters can be tuned to suit the specific requirements of the problem domain.

## 3.2 SELECTING RECOURSE PAIRS

Next, we select a set of $K \leq N^2$ pairs of recourse examples from $\mathcal{R}$ which will be presented to an LLM for cost comparison. This process can be understood as connecting the recourses into an undirected graph structure. In forming this graph, we enforce that each recourse must have a minimum of $K_{\min}$ edges, and that the graph as a whole forms a single connected component (where a path exists between all pairs of recourses). We find that this improves the performance of the final cost function, as it allows the costs for all recourses to be estimated on the same scale. To enforce the relative cost criterion, we prioritise edges between recourses which perturb the same continuous feature at two different parts of the distribution by exactly the same amount. This has the effect of forcing the LLM to reason about the difference in cost between e.g. increasing salary from 30-35k versus 50-55k. We also add edges to enforce comparisons of the same feature changes for different feature dependencies, e.g. two recourses which have the same increase in loan amount, but different credit ratings, which can be used to enforce the relative cost criterion of Desideratum 3 (see Section 4 later). The total additional edges from this enforcement is set to 10% of the total data for both, adding 20% extra data on average. Aside from these considerations, we find that the algorithm used to construct the graph of recourse pairs is relatively unimportant. In practice, any algorithm forming a connected graph subject to the $K_{\min}$ constraint seems to work well. We used a random spanning tree algorithm in all experiments.

## 3.3 PAIRWISE LLM LABELLING

For the standard prompt structure $\mathcal{B}$ (see Appendix D), we begin by instructing the LLM that it is a helpful assistant to a data scientist which labels data. It is then told the task of comparing two

individuals and their respective feature changes. We then enumerate the features, as well as their descriptions. The LLM is then asked to reason about which of the two given recourses requires more effort for the individual to achieve (i.e. cost), and finally to respond with a label of $1$ (first requires more effort), or $0$ (second requires more effort). Optionally, we also permit a third category of $0.5$, indicating a judgement that equal effort is required, which is a useful de-biasing signal in contexts where features represent sensitive demographic attributes. The prompt then gives a high-level overview of the desiderata in Section 2. In addition, the LLM is instructed to use chain-of-thought to increase performance and reduce social biases (Kamruzzaman & Kim, 2024). In other experiments, we also fine-tune the prompt more with a set of desired cost function parameters, denoted by $\mathcal{B}'$. For the full prompts, see Appendix D. The output of this stage is a set of $K$ comparisons $\mathcal{Q} = \{(i, j, y)\}_{k=1}^{K}$, where $i$ and $j \neq i$ are indices of a pair of recourse examples from $\mathcal{R}$ and $y \in \{0, 0.5, 1\}$ denotes the LLM's effort/cost judgement.

## 3.4 TRAINING THE COST FUNCTION

Finally, we use the dataset of LLM comparisons $\mathcal{Q}$ to train a cost function $C : \mathbb{R}^d \times \mathbb{R}^d \to \mathbb{R}_{\geq 0}$. Inspired by Rawal & Lakkaraju (2020), as well as the dominant approach to learning reward models from pairwise comparisons (Kwon et al., 2023), we train cost functions using the Bradley-Terry model. That is, given a cost function $C$ and a pair of recourses $(x_i, x_i')$ and $(x_j, x_j')$, we define the predicted probability that recourse $i$ has higher cost than recourse $j$ as

$$\hat{y}_C(i, j) = \frac{1}{1 + \exp(C(x_j, x_j') - C(x_i, x_i'))}. \tag{2}$$

Our cost function training objective is to minimise the binary cross-entropy between these predicted comparison probabilities and the labels provided by the LLM across all training examples:

$$\arg\min_{C \in \mathcal{M}} \left[ - \sum_{(i,j,y) \in \mathcal{Q}} y \log(\hat{y}_C(i, j)) + (1 - y) \log(1 - \hat{y}_C(i, j)) \right], \tag{3}$$

where $\mathcal{M}$ is a chosen model class. Since this loss is differentiable, we can define $\mathcal{M}$ as the class of MLP neural networks and train by stochastic gradient descent. As an alternative, we also consider the class of axis-aligned decision trees up to a maximum leaf count $L_{\max}$, which offers greater transparency. To train a non-differentiable tree with the pairwise Bradley-Terry loss, we use a bespoke algorithm developed by (Bewley & Lecue, 2022) (and refined in (Bewley et al., 2022)).

We one-hot encode categorical features (or binary encode ones with only 2 categories) and concatenate the original data point $x$, the perturbed recourse point $x'$ and the feature-wise difference $x' - x$ into a single vector $[x, x', x' - x] \in \mathbb{R}^{3d}$. In practice, we found that this simple feature augmentation step significantly improved the models' ability to learn costs. As a final post-processing step, we shift the outputs of trained models to $\geq 0$ on all training data. This has no impact on the Bradley-Terry loss, but produces the expected behaviour for a non-negative cost function to only output non-negative values.

## 4 EVALUATION

In our evaluation, we seek to understand how to train effective cost functions utilising LLMs. Throughout, we focus on three datasets, the Home Equity Line of Credit (HELOC) dataset (Mstz, 2024) for predicting whether someone will repay their account, the Adult Census dataset (Becker & Kohavi, 1996), for predicting if an individual earns higher than 50k per year, and the German Credit dataset (Hofmann, 1994), for classifying a client's credit risk. All are binary classification tasks, and we considered the first 800 instances from each dataset for training/testing of the cost function. All categorical features were modeled as binary 0/1 options, except German Credit which has multi-categorical features one-hot encoded. After creating the dataset of pairwise comparisons, and adding the additional links described in Section 3, we had 22,000 pairwise training examples on average, which was divided into 80/20% training/testing, respectively, for the cost functions. Currently GPT-4o represents state-of-the-art performance on many benchmarks (OpenAI, 2024), and indeed it is shown to be fairer than prior models Bowen III et al. (2024), so we used it in all our tests. As our data is mostly synthetic, we do not expect GPT-4o's known memorization of the datasets to be an issue (Bordt et al., 2024).

## 4.1 COMPARING HUMAN AND LLM JUDGEMENT OF COST

A natural first question is whether or not LLMs can provide a judgement of cost that aligns with human intuition. Hence, our evaluation started with a study to compare pairwise choices of cost between GPT-4o and humans. Participants were shown two individuals, a proposed change (i.e. their recourse), and asked to select which of these would require more "effort" (i.e, cost). We limited the options to a forced choice between Recourse 1 and Recourse 2, with no equal effort option, which we primarily reserved for situations involving demographic fairness (which this study did not involve). With a distribution of responses from humans in hand, this was then compared to GPT-4o's responses on the same questions using our prompt template in Appendix D. Note that because the LLM would arbitrarily choose Option 1 when its chain of thought communicated it was unsure, we allowed it a third option to identify this, and then replaced these data points with random answers. This allowed a more accurate comparison to humans, who tend to choose at random when unsure (Gigerenzer & Goldstein, 1996).

The materials covered all three datasets, with six questions for each. These six questions were split into three sets of two representing the first three parts of the desiderata, respectively.[2] Participants were also asked to choose how "close" they felt the two were, so we could compare their uncertainty with the LLM. See Figure 6 for an example and the supplementary material for the full survey.

We randomly recruited thirty industry data scientists for the purposes of the study. The participants were not compensated; all volunteered to participate. In total, 20 of the participants were male, 10 were female, all were aged 18+, and there was a mix of native/non-native English speakers.[3] The study obtained IRB approval.

The metric of interest was how the distributions of responses from humans matches that of the LLM. The test used was the Chi-square test of independence. A second metric was whether or not the most common response from the LLM and humans was identical, represented as mode: Y (they were equal), or mode: N (they were not equal). Lastly, we asked humans to quantify how far apart they felt the two options were, so we could quantify their certainty compared to the LLM.

The LLM was unsure of the answer 13.8% of the time, and this was replaced with random responses to simulate human uncertainty. Correlating LLM uncertainty to humans, we observe a strong positive correlation (Person's $r$=0.5; $p < 0.04$) in Figure 7, indicating that users and the LLM had approximately the same level of uncertainty across the same questions. Figure 2 shows more results. Overall, there is a tendency of the LLM to accurately align with the human labellers, with 15/18 of questions having statistically similar distributions (i.e. $p > 0.05$). When considering the most common responses (i.e. mode: Y), 15/18 different questions are also in agreement, two of

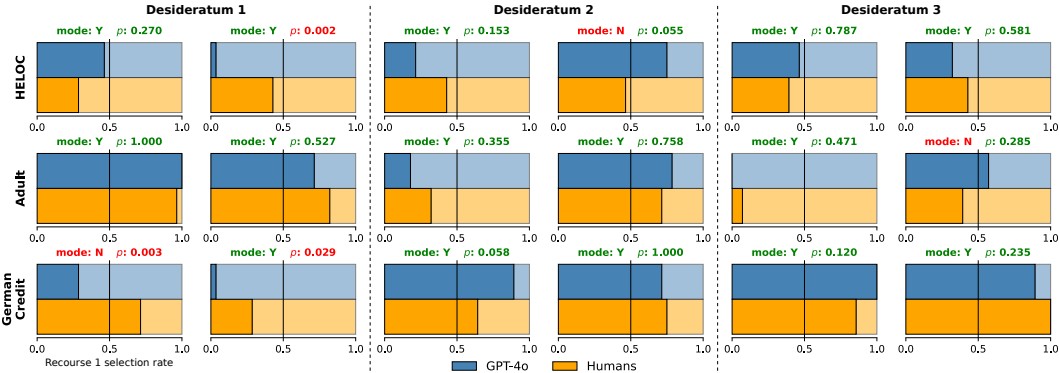

Figure 2: Human Study Results: 15/18 of the questions had the same modal response (i.e. the mode was Y), and 15/18 statistically similar distributions. Overall, 17/18 had one or the other, and were mostly aligned. Note we are trying to show the $p$-value is greater than 0.05, because we do not want to reject the null that humans and LLMs are aligned in cost judgement.

---

[2]Note that we did not evaluate Desideratum 4 with humans due to ethical concerns.

[3]Although the human sample is somewhat biased, results show they are aligned with the LLM, which increasingly simulate population user responses in surveys (De Bona et al., 2024).

these encompassing the questions without statistically similar distributions. Together, this can be interpreted to suggest that the LLM is in alignment for 17/18 of the questions. These results highlight that LLMs largely agree with human judgment of cost.

## 4.2 Training the Cost Functions

We used two types of prompts to label data for the cost functions, the *standard prompt*, and the *custom prompt*. The standard prompt is identical to what was used in the user study and uses only a high-level description of the desiderata to instruct the LLM, whilst the custom prompt attempts to fine-tune the resultant cost function with a ground truth we defined in the prompt (i.e., $B'$ in Figure 1). The point of this custom prompt is to see if we can e.g. re-order feature importance, manipulate the spectrum of cost for numerical features, add dependencies, and fairness attributes, see Section H for details on the ground truth chosen. This is important because (for example) the definition of fairness varies (Mehrabi et al., 2021), so we need to fine-tune different aspects of the cost function in practice. The choice of ground truth is largely irrelevant, we are simply seeing if it can be worked into the final cost function via the prompt. We trained either an MLP model or a tree for 50,000 batches of size 32. So, in total, there are 2 models we are testing across 3 datasets with 2 prompt types. We chose these models because trees help with transparency required in financial applications (Bewley et al.), and MLPs are differentiable, which is often required in recourse algorithms (Wachter et al., 2017).

## 4.3 Dependency Test

Perhaps the primary advantage of using LLMs to learn cost functions is that they have the potential to naturally model causal feature dependencies, which is the most intractable part of hand-designing a cost function. In this test, we examine the ability of LLMs to naturally label this with our standard prompt (i.e., no dependencies are mentioned in the prompt). We consider both synthetic and real data in this process. Synthetic data is considered because there is a risk that the LLM can only reason about causal dependencies on well known recourse datasets used for counterfactual generation, as the generated counterfactuals may be in the LLM training data.

**Synthetic Data** The generation of the synthetic data is detailed in Appendix K. In short, we crafted a novel dataset of known scientific dependencies in a medical domain, where data privacy laws should give additional reassurances that no such dataset was used to fine-tune the LLM, or subsequently used in recourse research papers. The dependencies where (1) that it is harder to lower cholesterol levels with a high saturated fat intake, (2) that it is harder to lower blood pressure with a high dietary salt intake, and (3) that it is harder to lose weight if consuming a large amount of heavily processed food. The ground truth was always Recourse 2, and we allowed the LLM to chose Recourse 1 or 2 as the one of higher cost, or 0 (i.e., uncertain) . We compared our standard prompt to an ablated version without the desiderata, to understand more how this helps. The most important difference between these is that the ablated prompt has no explicit instruction to consider dependen-

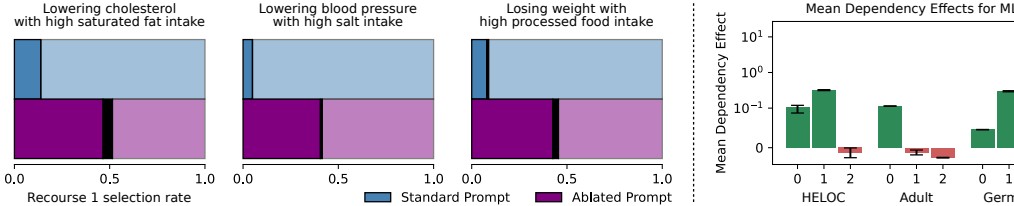

Figure 3: **(left) Synthetic Data**: Comparing the standard prompt (with the desiderata included in the prompt) and the ablated version (without the desiderata), the LLM was 90% + accurate at labelling the three known scientific causal dependencies, but only with the deisderata inserted into the prompt. Note the black areas in the data indicate the probability of the LLM being uncertain. **(right) Real Data:** The trained MLP cost functions successfully learned 6/9 of the ground truth dependencies suggested by Claude Sonnet 3.5., showing a general trend that the LLM can generally identify suitable dependencies in its labeling which are subsequently learned by the cost functions.

cies when evaluating cost of recourses. The results are shown in Figure 3(left), where the standard prompt correctly identified all three dependencies with a mean accuracy of 91%, compared to the same prompt with the desiderata ablated which was not significantly better than random guessing. Overall, this shows how we can trust the LLM to label reasonable causal dependencies in novel domains, but only if we (1) use chain-of-thought prompting and (2) the desiderata[4], which includes instructions to the LLM to explicitly look for dependencies.

**Real Data.** We prompted Claude Sonnet 3.5 to list the most important feature dependencies in each dataset (to help avoid leakage with GPT-4o), and repeated this 10 times to pick out three which were listed the most for our ground truths, see Appendix D and G. We iterated all the testing data with each cost function variation, and manually adjusted the data to subtract the cost of the less costly recourse option from the higher, hence, a positive score shows that the dependency is present in the cost function. In Figure 3 and Figure 4, we refer to this as the *"Mean Dependency Effect"*, where positive scores indicate the dependency has been learned to match the ground truth. The present results can be seen in Figure 3(right). Overall, 6/9 of all dependencies were modeled in accordance with Claude's ground truth in the MLP cost function, showing a generally positive ability to learn appropriate dependencies. In contrast, the tree models only learned one of these with the other eight showing 0 cost. The reason for this is likely that the tree would require most splits to learn the necessary dependency, but the MLP forms a smoother interpretation of the labels and learned the dependencies more easily.

## 4.4 FINE-TUNING EXPERIMENTS

Going forward, we consider a suite of experiments which try to fine-tune the prompt to achieve different results in the cost function. This is because judgement of cost often needs to be tuned to certain context. For example, during an economic downturn, a bank might need to adjust its lending criteria and weight features differently. In addition, the definition of fairness varies substantially between contexts (Mehrabi et al., 2021), so this also needs to be finetuned occasionally. We design a fine-tuning experiment using our custom prompting scheme $B'$ originally described in Figure 1. The prompting scheme adds a high-level description $B'$ to the original prompt $B$ indicating (1) how costly each feature should be to mutate in order, (2) how numerical features should change in cost at different parts of the distribution, (3) any dependencies we want to exaggerate or create, and (4) any fairness attributes desired. Not all these need to be specified during fine-tuning of cost functions, but we test all here for a complete experiment. When perturbing features in the subsequent

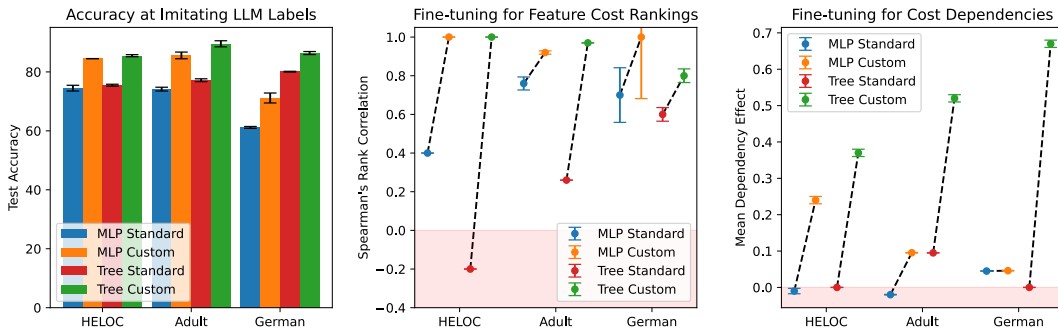

Figure 4: Accuracy and Desideratum 1 and 3 Fine-Tuning: (left) Accuracy of the cost functions at imitating the LLM's pairwise labels on test data. (middle) Ability of the cost functions to be fine-tuned to correlate with ground truth feature cost rankings specified in the custom prompt. A score of 1 illustrates the rankings are perfectly learned by the cost function. (right) Ability of the cost functions to be fine-tuned to weight dependencies specified in the custom prompt. Notably, the Adult MLP flips from a negative to positive dependency effect, showing we can reverse the cost of certain dependencies if desired. Standard error is shown. Red background indicated negative correlation or dependency effect for middle and right plots, respectively.

---

[4]In additional unreported experiments, we found that the ability of the LLM to reason successfully about causal dependencies requires chain-of-thought prompting.

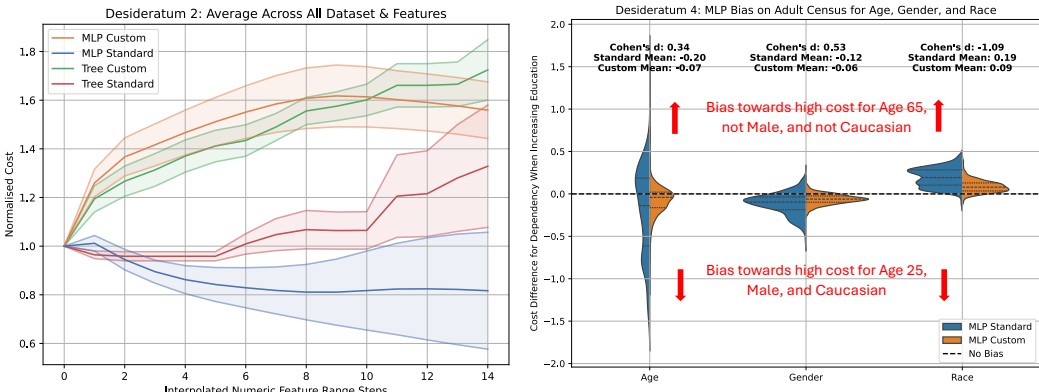

Figure 5: Desideratum 2/4 Fine-Tuning Results: (left) As the numeric features rise in value for the custom models, so does their respective cost relative to the standard ones. (right) Instructing the LLM to not unfairly discriminate between demographic information increased its fairness when suggesting recourse which increased education level. Overall, these results show how it is possible to fine-tune the cost function on Desideratum 2 and 4.

tests, categorical features were either flipped for binary or randomly changed for multi-categorical features, numerical features were perturbed upwards a standard deviation.

**Desideratum 1** Here the ground truth specified in the custom prompt was a specific rank ordering of how costly each feature should be to mutate. Each testing datum had each feature perturbed to test its cost, the results for each feature were averaged and reported across four random seeds. Each feature was rank ordered in a list and compared against the ground truth defined in the custom prompt with Spearman's rho $\rho$. Results can be seen in Figure 4(middle), where the custom prompt is compared to the standard one (which did not specify what the most costly features should be). Overall, the custom prompt-based cost functions successfully moved towards the new features rank orderings as instructed in $\mathcal{B}'$, with Heloc learning them perfectly, illustrating that it is possible to realign the relative importance of features if desired.

**Desideratum 2** Here the ground truth specified in the custom prompt was that each numerical feature should be harder to mutate the higher it gets in value. Each testing datum had each numerical feature perturbed upwards to test its cost at 16 evenly spaced intervals. Each feature across all datasets were averaged and again shown across four random seeds in Figure 5(left). The average Spearman's $\rho$ for the custom models across all features and datasets was 0.41, compared to 0.04 on the standard models, showing that the numerical features have a gradual trend of increasing their cost the higher the mutation starts, which aligns with the original ground truth schematic $\mathcal{B}'$ given to the LLM. This illustrates that it is possible to fine-tune the relative cost of numerical features across their spectrum if desired.

**Desideratum 3** Here the ground truth specified in the custom prompt was to purposefully enforce the worst performing previously tested dependencies in each dataset in Section **??**(right). The point is to see if we can correct them to be a positive mean cost dependency. As before, each testing datum had each dependency tested the same as Section 4.3, all were averaged and again shown across four random seeds in Figure 4. Notably, the negative cost associated with the dependency in Adult Census and Heloc flipped to be positive, showing it is possible to fine-tune this if desired. Moreover, the tree models all went from no/little cost associated with each dependency, to a positive one. Lastly, the strength of the positive cost in Heloc and German Credit for the MLP models increased, showing that by adding the dependency directly to the prompt, we can strengthen the dependency cost. This illustrates that it is possible to fine-tune dependencies if desired, simply by instructing the LLM to explicitly consider this dependency.

**Desideratum 4** Here the ground truth specified in the custom prompt was that the LLM should never use demographic information when considering the cost of other mutations, so here we tested

if mutating education upwards differed in cost between demographics. Each testing datum in Adult Census and had its education perturbed upwards while considering the datum being male/female, white/not-white, and aged 25/65. Figure 5 shows the results were the custom prompting with these fairness constraints was significantly less biased than the standard prompt alternative in all three demographic features. Specifically, Cohen's d was 0.34, 0.53, and -1.09 for age, gender, and race, respectfully, showing small to large effect sizes. This illustrates that it is possible to make the cost function fairer simply by adding this constraint to the prompt.

## 4.5 COST FUNCTION FIDELITY

It's important to understand how accurate the decision tree and MLP cost functions are at imitating the LLM's reasoning, since we are trying to distill the LLM's knowledge into small cost function, which can be judged based on how accurately it predicts pairwise comparisons the LLM labeled. Note there is noise in the LLM labels due to its inherent temperature settings, so 100% accuracy would be unwarranted, and indeed some noise has been shown to improve preference learning (Laidlaw & Russell, 2021). Models were trained for 50,000 batches of size 32, and evaluated on the labels of the remaining pairwise comparisons labeled by the LLM described in Section 4. The results can be seen in Figure 4(left). Overall, the custom models always achieved higher accuracy, because there was less noise in their labeling process due to the specific constraints in the prompt. Tree models on average also did better, but this is mostly due to their ability to classify equal cost, which the MLP could not, as it has a non-discrete function output. German Credit performed worse on average also due to the sparser one-hot encoding feature space. [5] Overall, this illustrates that the cost functions have learned to imitate the original LLM labeling well.

## 4.6 CASE STUDY

Here we showcase how our cost functions can improve current recourse algorithms by Keane & Smyth (2020) and Wachter et al. (2017), the prior being a data driven approach with an $L_1$ cost function and the latter a gradient-based method using a median-absolute deviation (MAD) cost function. A simple MLP classifier was trained on Adult Census and achieved 82% accuracy on the training and testing data. Note we repeated this evaluation on Heloc and German Credit in Appendix I. The data was standard normalised for a fair comparison between features when checking distances using each method's default cost function, this was then compared to our custom MLP cost function which was plugged into each method. Adult census was used in all tests with the standard prompting scheme. For full implementation details of each method, the data, and the architectures see Appendix I. In total, we evaluated on the same 6000 instances for each method, and for each of these we attempted recourse generation if they were negative predictions by the model initially. Keane & Smyth (2020) generated 1027 successful recourses, whilst Wachter et al. (2017) was an average of 2322 between methods.

| | Male | Age | Native-US | Married | Education | Hours-Work | Private Work | Caucasian |
|---|---|---|---|---|---|---|---|---|
| **Keane and Smyth (2020)** - Data Driven | | | | | | | | |
| $L_1$ | 0 | 554 | 1 | 29 | 266 | 177 | 0 | 0 |
| Ours | 0 | 380 | 1 | 30 | 275 | 341 | 0 | 0 |
| **Wachter et al. (2017)** - SGD Driven | | | | | | | | |
| MAD | 20 | 53 | 78 | 332 | 1594 | 4 | 8 | 1 |
| Ours | 1 | 28 | 3 | 313 | 1750 | 174 | 84 | 0 |

Table 1: Case Study Results: Each number represents the number of times each method recommended mutating that feature for recourse. In Keane & Smyth (2020), our method recommended mutating age less and hours-worked more as the main trade-off. In Wachter et al. (2017), our method recommended mutating education and hours-work in comparison to MAD which favored features such as male, and native-US, which are generally considered less actionable.

---

[5]These two models were also compared against a baseline LLM GPT-4o which was instructed to label every recourse option with a numerical cost value from 0-1, but the results were not competitive and not reported.

Table 1 shows the results where we counted how often each method recommended mutating a particular feature to achieve recourse. Broadly speaking, from a feature weighting perspective, the method by Keane & Smyth (2020) prioritized age as a feature, whilst our method reduced this by focusing instead on hours-work. For the method by Wachter et al. (2017) the default MAD cost function suggested many questionable recourses such as changing gender 20x more than our cost function, age almost 2x times, and even race once. However, perhaps the most inactionable feature (native-us) was suggested 78 times, compared to ours which was just 3 times.

## 5 RELATED WORK

In the counterfactual literature, early work used the median absolute deviation as a distance function (Wachter et al., 2017), which has some desirable properties such as robustness to outliers, but can't deal with categorical features or actionability constraints. Early work in this area by Ustun et al. (2019) proposed total and maximum-log percentile shift measures, which can address *relative cost*, but not the other desiderata constraints. Other researchers such as Karimi et al. (2020) proposed a weighted combination of $L_p$ norms across features, which does deal with *feature cost*, but again misses the other constraints in Section 2. Other recent work continued the use of $L_p$ norms (Karimi et al., 2020; Ramakrishnan et al., 2020), while others investigated HCI questions (Tominaga et al., 2024). In a more recent trend, work has begun to focus on individualised cost (De Toni et al., 2022; Yetukuri et al.; Nguyen et al., 2024), with some also focusing on the Bradley-Terry loss for pairwise comparisons specifically (Rawal & Lakkaraju, 2024), albeit without LLM assistance. In comparison to these works, we are concerned with how to automate the training of high-performing cost functions at scale with LLMs, which follows all the core desiderata constraints in Section 2 laid out in the literature.

LLMs have recently been applied to various tasks (Han et al., 2024; Hollmann et al., 2024; Hegselmann et al., 2023; Borisov et al., 2022), but, here we are focused on utilising their latent knowledge for labeling data for training cost functions, which has not been explored before. Perhaps the most similar work to ours was suggested by Rawal & Lakkaraju (2020). Specifically, they learned a preference function using the Bradley-Terry Loss, pairwise comparisons, and MAP estimates (Hunter, 2004; Caron & Doucet, 2012). However, their approach would require human labellers, and doesn't take *relative* or *dependent* feature cost into account. To help automate similar processes in related areas, recent work has utilised LLMs as judges or evaluators to produce pairwise preferences for learning reward models. Most popularly they are used in RLHF for aligning language models with human preferences (Ouyang et al., 2022), but the ability of this to help with cost functions has not been evaluated until the present work.

There is a literature on evaluating how well LLMs correlate with human judgement, but it is difficult to interpret because as much work has shown positive results (Liu et al., 2023; Chiang et al., 2024), as negative (Bavaresco et al., 2024; Koo et al., 2023). Some of this work has highlighted how the discrepancy of results is likely due to a narrow focus on tasks (Bavaresco et al., 2024), suggesting that LLMs may need to be evaluated on very specific use cases to uncover credible ones. Bearing this in mind, the ability of LLMs to correlate with human judgement of cost has not been explored previously, which we addressed in the present paper with our human study.

## 6 CONCLUSION

The problem of algorithmic recourse, and counterfactual explanation more broadly, has grown in importance the past several years as AI is increasingly used for high-stakes decisions (Keane et al., 2021; Karimi et al., 2022; Gajcin & Dusparic, 2024; Kothari et al., 2024). However, one of the core unresolved issues plaguing research in the area has been the lack of appropriate cost functions, which has limited the practical value of recourse recommendations. In this paper, we first explored LLM's natural ability to align with human judgments of cost, showing that they do largely correlate. We then showed that the cost functions can be fine-tuned to fit a variety of use cases. Lastly, we also demonstrated the practical outcomes of using these cost functions in two real-world algorithms. In future work, it would be interesting to investigate the ability of LLMs to learn gain functions for semifactual recourse (Kenny & Huang, 2024), as opposed to counterfactual recourse, which likely involves other considerations.

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

Jayanth Yetukuri, Ian Hardy, and Yang Liu. Actionable recourse guided by user preference.

## A    ACTIONABILITY CONSTRAINTS AND FEATURES USED

The following are the datasets and features used, alongside any actionablity constraints employed throughout the paper:

**HELOC Dataset**: Here, the actionability constraints were to clamp feature mutations at the highest and lowest values observed in the dataset.

- *MSinceMostRecentInqexcl7days*: Number of months passed since the last credit inquiry on the individual.
- *NumRevolvingTradesWBalance*: The number of the individual's current credit accounts (e.g. credit cards) that have balances on them.
- *NumTradesOpeninLast12M*: The number of new credit accounts opened in the last 12 months.
- *NumInqLast6M*: The number of credit inquiries carried out on the individual in the last 6 months.

**Adult Census Dataset**: Here, the actionability constraints were to clamp feature mutations at the highest and lowest values observed in the dataset. Also, age and education number were only allowed to move upwards.

- *isMale*: If the person is male, or female, represented as 1 or 0, respectively.
- age: The person's age, represented as a floating point number.
- *native-country-United-States*: If the person's birthplace is the United States, or not, represented as 1 or 0, respectively.

- *marital-status-Married*: If the person is married, or not, represented as 1 or 0, respectively.
- *education-num*: The person's level of education, represented by a positive integer, where higher numbers are higher levels of education.
- *hours-per-week*: The number of hours the person works per week, represented by a positive integer.
- *workclass-Private*: If the person works for a private company, or is self-employed, represented as 1 or 0, respectively.
- *isCaucasian*: Is the person white or not, represented as 1 or 0, respectively.

**German Credit Dataset**: Here, the actionability constraints were to clamp numeric feature mutations at the highest and lowest values observed in the dataset.

- *status*: Status of existing checking account.
- *duration*: The proposed duration of the loan in months, expressed as an integer.
- *credit history*: The person's credit history with the options.
- *purpose*: The purpose of the loan.
- *amount*: The size of the loan asked for.

## B  PERTURBATION FUNCTION

In the context of feature vector perturbation, we employ a probabilistic approach to introduce controlled mutations to the feature set. Specifically, we perturb a feature vector by altering a random subset of its components. The number of features to be perturbed, denoted as $k$, is selected from the discrete set $\{1, 2, 3, 4\}$ with a predefined probability distribution. The probability mass function (PMF) for $k$ is given by:

$$P(K = k) = \begin{cases} 0.8 & \text{if } k = 1 \\ 0.1 & \text{if } k = 2 \\ 0.05 & \text{if } k = 3 \\ 0.05 & \text{if } k = 4 \end{cases}$$

This distribution ensures that perturbing a single feature is the most probable event, while perturbing four features is the least probable. The purpose was to focus on sparsity for the cost function training, but also have some robustness. When perturbing numeric features, they have four possible values in our tests. All numeric features can be perturbed upwards one standard deviation, or half a standard deviation. If actionability constraints allow, they can also be perturbed down the same two values, they were then rounded to the nearest integer.

## C  OUT OF DISTRIBUTION EXPERIMENT

In addition, we were interested in how our cost functions, which were trained to specialise in sparse single feature modifications performed out of distribution when scoring multiple feature mutations in recourse. Hence, we trained a custom tree and MLP model on Adult Census with only 2 feature perturbations allowed, which achieved 88.2% and 87.5% test accuracy on the LLM labels, respectively, and dropped by 82.1% and 81%, respectively. This drop in performance constituted an average of 6.15%, and shows that performance is largely maintained out of distribution, but for maximum effect the training data should represent what is desired in deployment.

## D  PROMPTS

Here are the prompts for HELOC, all other datasets followed the exact same pattern, and all can be seen in the code base if desired. Note in the actual prompts we instructed the LLM to use 1, 2, 0,

to select Recourse 1, Recourse 2, and neither, respectively, although in the main paper we used 1, 0, and 0.5, as this more accurately reflected the Bradley-Terry model.

Standard prompt ($\mathcal{B}$):

```
You are a helpful assistant to a data scientist to help them
label data.  You will be shown a data point representing a person
Alex, and a mutation of it, You will also be shown a data point
representing a person Jaden, and a mutation of it, your task is
to label which of the two mutations would take more effort to
achieve.

The data will be the HELOC Dataset which uses these features:

MSinceMostRecentInqexcl7days:  Number of months passed
since the last credit inquiry on the individual.
NumRevolvingTradesWBalance:  The number of the individual's
current credit accounts (e.g.  credit cards) that have balances on
them.  NumTradesOpeninLast12M: The number of new credit accounts
opened in the last 12 months.  NumInqLast6M: The number of credit
inquiries carried out on the individual in the last 6 months.

The data is represented in array form like ['MSinceMostRecentInqexcl7days',
'NumRevolvingTradesWBalance', 'NumTradesOpeninLast12M',
'NumInqLast6M']

Now consider the following individual Alex:  """+str(x1)+""" Now
consider this mutation of Alex:  """+str(x1p)+"""

Now consider the following individual Jaden:  """+str(x2)+""" Now
consider this mutation of Jaden:  """+str(x2p)+"""

Which of these two mutations would take more effort?  You must
provide an answer.
```

```
Remember the following 4 rules and use them in your decision:

1.  Some features are naturally harder to change than others, use
this logic.

2.  For numerical features, the difficulty of changing them can
often depend on their starting values.

3.  Apart from the mutated features, consider the other features
which are different between Alex and Jaden, and how this may
affect difficulty.

4.  Do not ever use demographic features (e.g., age, gender, race)
when considering the difficulty of mutating other features.
```

```
Outline your reasoning process step by step, before giving your
answer as 1, 2, or 0 in the tags <answer>...</answer>, where 1
means you think the first mutation requires more effort, 2 means
you think the second mutation requires more effort, and 0 means
you think there is no difference.
```

Custom Prompt ($\mathcal{B} + \mathcal{B}'$):

```
You are a helpful assistant to a data scientist to help them
label data.  You will be shown a data point representing a person
Alex, and a mutation of it, You will also be shown a data point
representing a person Jaden, and a mutation of it, your task is
to label which of the two mutations would take more effort to
achieve.

The data will be the HELOC Dataset which uses these features:
```

MSinceMostRecentInqexcl7days:  Number of months passed
since the last credit inquiry on the individual.
NumRevolvingTradesWBalance:  The number of the individual's
current credit accounts (e.g.  credit cards) that have balances on
them.  NumTradesOpeninLast12M: The number of new credit accounts
opened in the last 12 months.  NumInqLast6M: The number of credit
inquiries carried out on the individual in the last 6 months.

The data is represented in array form like ['MSinceMostRecentInqexcl7days',
'NumRevolvingTradesWBalance', 'NumTradesOpeninLast12M',
'NumInqLast6M']

Now consider the following individual Alex:  """+str(x1)+""" Now
consider this mutation of Alex:  """+str(x1p)+"""

Now consider the following individual Jaden:  """+str(x2)+""" Now
consider this mutation of Jaden:  """+str(x2p)+"""

Which of these two mutations would take more effort?

Remember the following:

1.  The hardest features to change, in order from the
hardest to easiest are [MSinceMostRecentInqexcl7days,
NumRevolvingTradesWBalance, NumTradesOpeninLast12M, NumInqLast6M]

2.  For the numerical features, they are all harder to increase
the higher they get.

3.  If NumInqLast6M is greater than zero, then increasing
'NumTradesOpeninLast12M' becomes more difficult.

Outline your reasoning process step by step, before giving your
answer as 1, 2, or 0 in the tags <answer>...</answer>, where 1
means you think the first mutation requires more effort, 2 means
you think the second mutation requires more effort, and 0 means
you think there is no difference.

Prompt to elicit numerical response from LLM:

You are a helpful assistant to a data scientist that helps them
label data.  You will be shown a data point representing a person
Alex, and a mutation of it.  your task is to label how much effort
this mutation was to achieve using a number between 0 and 1, where
0 is no effort, and 1 is the most possible effort.

The data will be the HELOC Dataset which uses these features:

MSinceMostRecentInqexcl7days:  Number of months passed
since the last credit inquiry on the individual.
NumRevolvingTradesWBalance:  The number of the individual's
current credit accounts (e.g.  credit cards) that have balances on
them.  NumTradesOpeninLast12M: The number of new credit accounts
opened in the last 12 months.  NumInqLast6M: The number of credit
inquiries carried out on the individual in the last 6 months.

The data is represented in array form like ['MSinceMostRecentInqexcl7days',
'NumRevolvingTradesWBalance', 'NumTradesOpeninLast12M',
'NumInqLast6M']

Now consider the following individual Alex:  """+str(x1)+""" Now
consider this mutation of Alex:  """+str(x1p)+"""

Using a floating point number between 0 and 1, how much effort was
this to achieve?  You must provide an answer.

Outline your reasoning process step by step before giving your answer in the tags <answer>...</answer>

Human Study Prompt:

You are a helpful assistant to a data scientist to help them label data. You will be shown a data point representing a person Alex, and a mutation of it, You will also be shown a data point representing a person Jaden, and a mutation of it, your task is to label which of the two mutations would take more effort to achieve.

The data will be the HELOC Dataset which uses these features:

MSinceMostRecentInqexcl7days: Number of months passed since the last credit inquiry on the individual. NumRevolvingTradesWBalance: The number of the individual's current credit accounts (e.g. credit cards) that have balances on them. NumTradesOpeninLast12M: The number of new credit accounts opened in the last 12 months. NumInqLast6M: The number of credit inquiries carried out on the individual in the last 6 months.

The data is represented in array form like ['MSinceMostRecentInqexcl7days', 'NumRevolvingTradesWBalance', 'NumTradesOpeninLast12M', 'NumInqLast6M']

Now consider the following individual Alex: """+str(x1)+""" Now consider this mutation of Alex: """+str(x1p)+"""

Now consider the following individual Jaden: """+str(x2)+""" Now consider this mutation of Jaden: """+str(x2p)+"""

Which of these two mutations would take more effort? You must provide an answer.

Remember the following 4 rules and use them in your decision:

1. Some features are naturally harder to change than others, use this logic.

2. For numerical features, the difficulty of changing them can often depend on their starting values.

3. Apart from the mutated features, consider the other features which are different between Alex and Jaden, and how this may affect difficulty.

4. Do not ever use demographic features (e.g., age, gender, race) when considering the difficulty of mutating other features.

Outline your reasoning process step by step, before giving your answer as 1, 2, or 0 in the tags <answer>...</answer>, where 1 means you think the first mutation requires more effort, 2 means you think the second mutation requires more effort, and 0 means you think there is no difference.

Prompt to acquire ground truth dependencies:

... (insert the previous standard prompt here) ...

In the above problem, what are the primary feature dependencies that may effect effort?

# E   HUMAN STUDY QUESTION EXAMPLE

Here we supply an example question from the human study for the HELOC dataset. The full survey can be seen in the supplement.

### Perceived Effort Required in Dataset Feature Mutations

**HELOC Dataset**

*Data Description:*
**The FICO HELOC dataset contains anonymized information about home equity line of credit (HELOC) applications made by real homeowners. The customers in this dataset have requested a credit line in the range of USD 5,000 - 150,000.**

*Selected Features:*
**1. Months Since Recent Inquiries: Number of months passed since the last credit inquiry on the individual.**
**2. Number of Credit Accounts with Balances: The number of the individual's current credit accounts (e.g. credit cards) that have balances on them**
**3. Number of New Credit Accounts: The number of new credit accounts opened in the last 12 months**
**4. Number of Inquiries: The number of credit inquiries carried out on the individual in the last 6 months**

\* 2.

| **Alex** | *Months since Recent Inquiry* | *Number of Credit Accounts with Balances* | *Number of New Credit Accounts* | *Number of Inquiries* |
|---|---|---|---|---|
| Features | 6 | 3 | 2 | 4 |
| Change(s) to Make | | | 4 | |

| **Jaden** | *Months since Recent Inquiry* | *Number of Credit Accounts with Balances* | *Number of New Credit Accounts* | *Number of Inquiries* |
|---|---|---|---|---|
| Features | 6 | 3 | 2 | 4 |
| Change(s) to Make | | 5 | | |

Which individual's proposed change would require more effort?

◯ Alex

◯ Jaden

\* 3. In your opinion, how much difference in effort do you perceive between the two changes (for Alex & Jaden) in the scenario above?

| 1 (Almost No Difference) | 7 (Really Large Difference) |
|---|---|
| ◯———————————————— | ▢ |

Figure 6:  Human study question example.

## F  UNCERTAINTY IN HUMAN STUDY

Here we consider the percentage of replies from the LLM in the Section 4 human study for each question which were uncertain (i.e., it chose the third option rather than Recourse 1 or 2), alongside the average response humans gave for the distance between the two recourse in Figure 6. For both lists, we normalized each to be between 0-1, and plotted them in a scatter plot to see the correlation. Both lists represent each group's uncertainty in choosing a recourse, and shows how the LLMs and humans correlate in this aspect to a high degree (Person's r=0.48; $p < 0.04$). What this tells us is that how uncertain humans an LLMs are on these recourse questions in the human study strongly correlate. Note that the correlation is identical with un-normalized scores also, we just do so here for clarity and visual purposes.

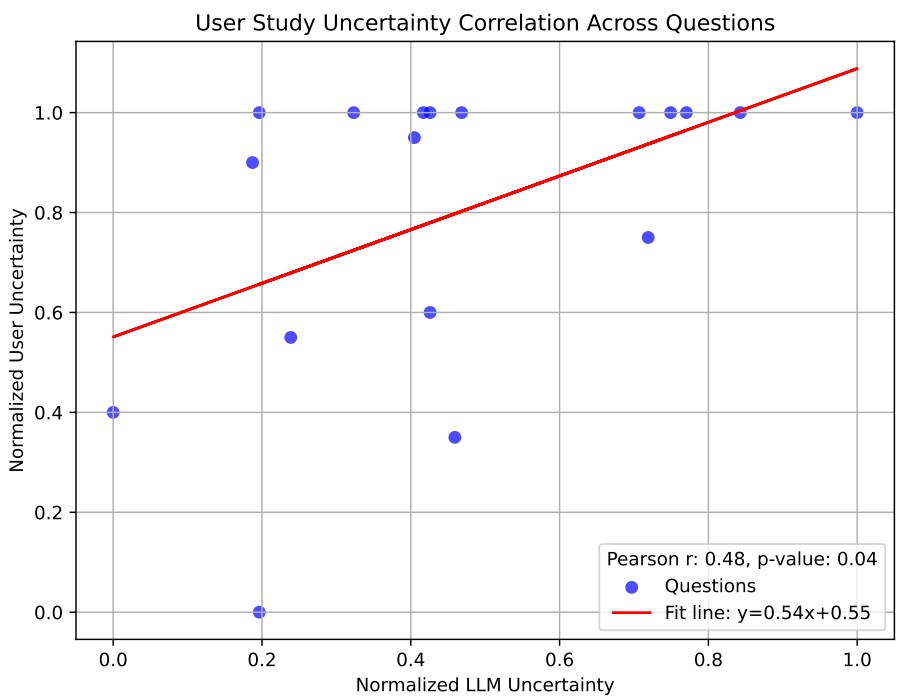

Figure 7: The correlation between LLM uncertainty and human uncertainty in the human study shows both groups were similarly uncertain on each question.

## G  FEATURE DEPENDENCIES

In Section 4.3 we evaluated how well various prompt types picked up on feature dependencies. To acquire these dependencies in an objective way, we queried Claude Sonnet 3.5 to list all feature dependencies in all 3 datasets using the standard prompt in Section D and adding:

...

```
If I change the same feature in Alex and Jaden the same amount,
but another feature is different which effects the effort
involved, what would be the most likely dependencies like this
to happen?
```

| Question | P-value | Same Most Common Response | LLM Uncertainty % |
|---|---|---|---|
| 1 | 0.27 | True | 0.00 |
| 2 | 0.00 | True | 0.00 |
| 3 | 0.26 | True | 7.14 |
| 4 | 0.18 | False | 42.86 |
| 5 | 1.00 | True | 28.57 |
| 6 | 1.00 | True | 46.43 |
| 7 | 1.00 | True | 0.00 |
| 8 | 0.53 | True | 0.00 |
| 9 | 0.35 | True | 0.00 |
| 10 | 0.53 | True | 3.57 |
| 11 | 0.47 | True | 0.00 |
| 12 | 0.59 | False | 71.43 |
| 13 | 0.00 | False | 0.00 |
| 14 | 0.03 | True | 0.00 |
| 15 | 0.06 | True | 0.00 |
| 16 | 0.56 | True | 32.14 |
| 17 | 0.12 | True | 0.00 |
| 18 | 0.24 | True | 17.86 |

Table 2: Results of 18 Question in Human Study: We are looking to see which have statistically similar distributions or the same most common response as a sign of LLM alignment with humans in judgement of cost. Overall, 17/18 show one metric or the other with positive results.

We repeated this 10 times and took the three dependencies which occurred most often, theses were:

HELOC:

1. If NumTradesOpeninLast12M is low, it makes it more challenging to increase NumRevolvingTradesWBalance.
2. If NumInqLast6M is high, it suggests it should be more difficult to increase NumTradesOpeninLast12M.
3. If NumInqLast6M is high, it should be more difficult to increase NumRevolvingTradesWBalance.

Adult:

1. Increasing working hours should be more difficult if you are married.
2. Changing marital status should be more difficult the older you are.
3. Increasing working hours should be more difficult if working for a private company.

German Credit:

1. A bad credit history should make it harder to increase your loan amount.
2. A bad credit history should make it harder to increase your duration.
3. The effort to decrease the duration of a loan should be harder for larger loan amount.

## H GROUND TRUTH FOR SECTION 4.4

We had to define a ground truth for our fine-tuning experiments to see how we could manipulate the four desiderata outlined previously. Note Desideratum 4 (i.e., fair cost) was only evaluate on Adult due to its numerous demographic features. The ground truth defined in $B'$ for each dataset was:

HELOC:

```
... 1.  The hardest features to change, in order from
the hardest to easiest are [MSinceMostRecentInqexcl7days,
NumRevolvingTradesWBalance, NumTradesOpeninLast12M, NumInqLast6M]
```

2. For the numerical features, they are all harder to increase
the higher they get.

3. If NumInqLast6M is greater than zero, then increasing
'NumTradesOpeninLast12M' becomes more difficult.

Adult Census:

... 1. The hardest features to change, in order from the hardest
to easiest are [native-country-United-States, isWhite, isMale,
age, marital-status-Married, education-num, workclass-Private,
hours-per-week]

2. For age, education-num, and hours-per-week, they are all
harder to increase the higher they get.

3. Increasing hours-per-week is more effort if the person works
for a private company.

4. Never use demographic information (i.e., isMale, age, isWhite)
when calculating the effort of other feature changes.

German Credit:

... 1. The hardest features to change, in order from the hardest
to easiest are [credit history, status, purpose, duration, amount]

2. For the numerical features, they are all harder to increase
the higher they get.

3. Having bad credit history or bad status makes it harder to
increase amount.

These dependencies where chosen to be fine-tuned because they performed badly in Section 4.3.

## I  BASELINE IMPLEMENTATION DETAILS

This section serves to give full details about the implementation of Keane & Smyth (2020) and
Wachter et al. (2017) in Section 4.6. The data used was Adult Census with 30,000 for training, and
6,000 for testing the recourse generation.

### I.1  KEANE AND SMYTH

This method is data driven and works by defining a case-base of recourse options for training
data (Keane & Smyth, 2020). In practice, each training data has its nearest unlike neighbor found in
the case-base and the difference between the two is taken as one recourse option. Recourses of 2 or
less feature changes are preferred by the authors, we focus on single feature changes. At test time, a
query has its nearest neighbour found in the case base and its recourse is applied to the query, this is
repeated for all nearest neighbours to find the best recourse option adhering to some constraints. For
us, these constraints are a single feature mutation, and that the result must be a valid counterfactual.
Finally, we also considered the 100 nearest neighbours as possible recourses.

### I.2  WACHTER ET AL.

A heavily implemented framework in research (Wachter et al., 2017), the method works by gener-
ating a set of random recourses which optimize to be closer to the query, while optimizing to also
be the counterfactual class. The second constraint is gradually up-weighted with a lambda term to
be more important throughout several optimization steps. We implement the method as normal with
300 possible counterfactuals during optimization, categorical features are snapped to the closest real
value, the results are filtered to those which are valid counterfactuals, and the closest chosen as the
answer. Because we are interested in sparse explanations, we also clamp each possible counterfac-
tual to have one possible feature mutation, which in practice is done allowing the largest currently
mutated feature to be the recommended recourse action.

## J    CASE-STUDY EXTRA RESULTS

To complete our case study in Section 4.6, we add the two other datasets in the paper. We focused on Adult Census in the main paper because it is less debatable what the most actionable features are.

| | MSinceMostRecentInqexcl7days | NumRevolvingTradesWBalance | NumTradesOpeninLast12M | NumInqLast6M |
|---|---|---|---|---|
| **Keane and Smyth (2020)** - Data Driven | | | | |
| $L_1$ | 336 | 9 | 1 | 0 |
| Ours | 270 | 43 | 8 | 25 |
| **Wachter et al. (2017)** - SGD Driven | | | | |
| MAD | 167 | 1 | 0 | 0 |
| Ours | 5 | 10 | 5 | 148 |

Table 3: Heloc Results: On average the baselines favored Months Since Most Recent Inquiry Excluding 17 days, in contrast to our cost function which favored Number of inquiries in the last 6 months and number of revolving trades with balance as a trade-off. Considering the first feature has a time constraint, it is immediately more actionable to modify the feature our cost function chose. Our cost function also generally offers a more diverse set of explanations.

| | Repayment Term | Loan Amount | Status | Credit History | Purpose |
|---|---|---|---|---|---|
| **Keane and Smyth (2020)** - Data Driven | | | | | |
| $L_1$ | 0 | 37 | 0 | 0 | 0 |
| Ours | 0 | 37 | 0 | 0 | 0 |
| **Wachter et al. (2017)** - SGD Driven | | | | | |
| MAD | 1 | 1 | 22 | 34 | 51 |
| Ours | 19 | 1 | 1 | 51 | 38 |

Table 4: German Credit Results: Keane and Smyth performed poorly on this dataset because (1) the dataset itself is smaller than the others (666 training), and is heavily imbalanced (95/5%), hence because it is a data driven method which directly uses the data for computation, there was sparse examples of how to generate counterfactuals. In Wachter et al. (2017) our method favored *Repayment Term* and *Credit History* in comparison to MAD which focused on *Status* and *Purpose*. Arguably, *Repayment Term* is the easiest feature to modify, as *Status* involves changing your savings amount which is quite costly when increasing, *Credit History* by comparison is easier to change but takes time, and *Purpose* which involves major changes to ones future plans.

## K    DEPENDENCY EXPERIMENT WITH SYNTHETIC DATA

Due to our datasets in the paper being popular recourse datasets, it is reasonable to assume that there are counterfactual pairs the LLM has seen during pre-training. Hence, to verify that it can detect causal dependencies without having seen direct examples from datasets, we create a synthetic dataset which does not exist anywhere, and thus cannot be part of the LLM's pre-training data.

We generated a medical dataset of personal information which is unlikely to have similar publicly available datasets used in recourse papers online. The features we used were:

- Saturated Fat Intake: the amount of saturated fat they eat, from 0 to 100 grams..
- Salt Intake: the amount of salt fat they eat, from 0 to 10 grams.
- Processed Food Intake: the amount of saturated fat they eat, from 0 to 500 grams.
- Cholesterol Level: Cholesterol level in mg/dl.
- Blood Pressure: their systolic blood pressure.
- Weight: weight in KG.

We chose these features because there are three scientifically known dependencies we can use as a ground truth.

1. It is harder to lower cholesterol if your saturated fat is too high.

2. It is harder to lower blood pressure if your salt intake is too high.

3. It is harder to lose weight if your intake of processed food is too high.

We generated this dataset according to reasonable values seen in the below script

```python
import numpy as np
import pandas as pd

np.random.seed(42)
n_samples = 1000

saturated_fat_intake = np.random.randint(0, 101, n_samples)  # 0 to 100 grams
salt_intake = np.random.randint(0, 11, n_samples)  # 0 to 10 grams
processed_food_intake = np.random.randint(0, 501, n_samples)  # 0 to 500 grams

cholesterol_level = np.round(200 + 0.5 * saturated_fat_intake +
    np.random.normal(0, 10, n_samples)).astype(int)
blood_pressure = np.round(120 + 2 *
    salt_intake + np.random.normal(0, 5, n_samples)).astype(int)
weight = np.round(70 + 0.1 * processed_food_intake +
    np.random.normal(0, 5, n_samples)).astype(int)

mortality_risk = ((cholesterol_level > 240) |
    (blood_pressure > 140) | (weight > 100)).astype(int)

data = pd.DataFrame({
    'Saturated Fat Intake': saturated_fat_intake,
    'Salt Intake': salt_intake,
    'Processed Food Intake': processed_food_intake,
    'Cholesterol Level': cholesterol_level,
    'Blood Pressure': blood_pressure,
    'Weight': weight,
    'Mortality Risk': mortality_risk
})
```

With this dataset in hand we iterated each datum 3 times and made the following 3 mutations each time to test each dependency. In each case the datum was duplicated to control for other features and focus only on the dependencies across a diverse range of data.

1. We set saturated fat to 10g v 100g, and took the action of lowering cholesterol by 10.

2. We set salt intake to 1g or 15g, and took the action of decreasing blood pressure by 10.

3. We set processed food intake to 10g v. 1500g, and took the action of losing 5KG of weight.

In all cases these mutations had additional random noise added to them for robustness. The LLM was allowed to answer that the first, second, or neither recourse was higher effort. In all cases, recourse 2 was the ground truth, hence, if most of the LLM responses are Recourse 2, then it is picking up on the causal dependencies. Lastly, we compared a standard prompting scheme with no information, and the same prompt with a high-level overview of the desiderata in Section 2. Results are shown in Figure 8. Overall, when adding the high-level desiderata to the prompt, the LLM can always detect these known causal dependencies with very high acccuracy. This shows the LLM is capable of reasoning about causal dependencies without being exposed to similar training data in the past. Moreover, what is particularly interesting is that by explicitly telling the LLM to consider

other dependencies (by adding the desiderata to the prompt), it is able to do that. However, without being told to consider dependencies in the prompt, it is not able to reason correctly.

In short, this experiment tells us two important things. First, LLMs can reason about causal dependencies it has not been exposed to before in terms of counterfactual data available on the internet. Secondly, in order to do this, the desiderata from Section 2 must be added to the prompt. Note, this is a general desiderata, not dataset specific, it does not make the method less general.

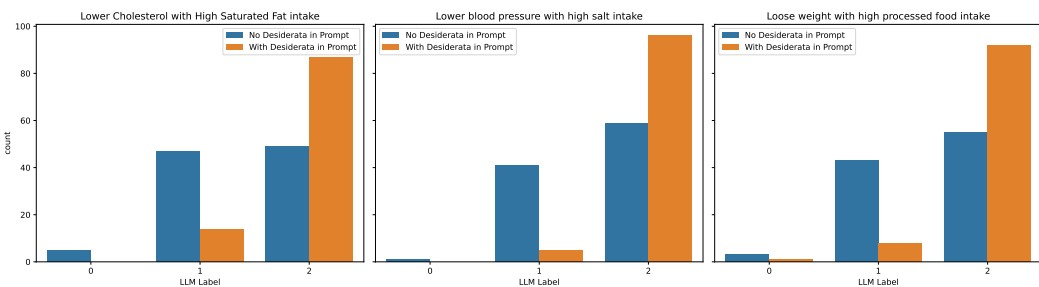

Figure 8: The labeling of the LLM on our synthetic causal relationships. The ground truth always corresponds to Recourse option 2. In general, the LLM was capable of modeling the causal dependencies with 90% accuracy. When ablating the desiderata from the prompt, this reduced to near random guessing between he two recourse options. Note, 0 corresponds to the LLM assigning equal cost to both recourses.

