# OpenReview forum: "Language Models Can Help to Learn High-Performing Cost Functions for Recourse"
_ICLR.cc/2025/Conference — Submitted to ICLR 2025_

### Official Review · Reviewer_8sk6 · 2024-10-31

**Soundness:** 3
**Presentation:** 4
**Contribution:** 2
**Rating:** 6
**Confidence:** 4

**Summary:**

This paper proposes to learn more realistic cost functions for algorithmic recourse (than commonly used L1/L2 costs) with the help of large language models (LLMs). The LLM's role is to assess pairs of two recourses and to decide which one is harder to implement. The ordering is then used to train an explicit cost function using learnable function approximators (e.g., trees, neural networks). The authors show that LLMs align with human judgements and background knowledge and that constraints the costs of feature manipulation can be explicitly added through the prompt.

**Strengths:**

Strengths:
* I think it is important to have more realistic cost functions for recourse and the problem of obtaining them is under-explored in the literature. This work contributes to this goal.
* Overall, the write-up is good to follow
* I do not have major concerns regarding soundness and see no major technical flaws in the described method

**Weaknesses:**

Weaknesses
* **User Study (Section 4.1.).** I have some concerns regarding the study:
    * I am not sure whether the independence test is a good measure of agreement. Independence would also not be obtained if the LLM responses and human responses are inversely correlated. In the social sciences, agreement measures such as Cohan’s kappa are established, and could be a suitable alternative to assess how high agreement beyond chance is between humans and the LLM. This scale also allows for a better interpretation of the values.
    * What exact test was used? The classical Chi2 test (https://en.wikipedia.org/wiki/Pearson%27s_chi-squared_test)  uses the null hypothesis of the observations being independent. Thus, rejecting the null hypothesis means we have sufficient evidence that they are dependent and small p values would imply dependency. However, in the Figure, it seems that large p values indicate dependency. This remains unclear to me and makes it hard to interpret the p-values. Please provide more details.
    * I am also not sure how the rejected pairs (where the LLM is uncertain) influence the score. How would the results change if scores where only the accepted pairs by the LLM are used were obtained?
    * Selection of examples: how exactly were the samples selected, how was made sure that they were realistic examples and of sufficient complexity (for straight-forward examples agreement would be obvious)? The number of two examples per dataset/desiderata seems rather low

* **Usefulness of incorporating constraints (Section 4.2).**
    * I am not sure what the advantage of using the LLM to incorporate constraints is over explicit methods like Ustun et al. (2019). Optimization-based procedures like this one allow to directly incorporate them and also come with guarantees. However, while the LLM seems to follow the constraints given, there are no guarantees. Thus, while it is interesting to see that we can incorporate these kinds of constraints it is not a unique advantage of this method and should potentially be compared with explicit recourse models that can handle constraints.

* **The LLM seems to fail to inherently model dependencies.** To me, the most important question is whether we can leverage implicit knowledge available in LLMs to speed up modeling of feature dependencies. Specification of all these causal relationships and mutual feature dependencies is what requires most effort when specifying cost functions, because (as mentioned) the effort for these grows exponentially with the number of features. Adding non-linear cost functions for a single feature (e.g., assigning a higher general weight to it, or making changes harder in certain ranges) is still feasible and already done in many works, for instance through quantile normalization and weighting (as mentioned, see Ustun et al., 2019 and Karimi et al., 2020). It would be much more interesting to test if the LLM can help model common causal relations, e.g., (adult dataset) it is extremely hard to increase the education from bachelors to PhD in 2 years or that marital status might be related to hours worked, when children have to be taken care of etc. This is the task where I think the LLM can save most manual effort. Unfortunately, the results regarding Desideratum 3 in Table 1 show that the LLM does not automatically take care of such dependencies when not explicitly specified. Therefore, I am unsure if the current approach does really reduce the effort, as  manual specification of all dependencies seems to be required still.
* **The application of the cost function in popular recourse frameworks is missing.** I wonder how the more complex cost function affects the efficiency of recourse frameworks and how easy it is to integrate. It seems in the case study, a brute-force approach to recourse was chosen. This makes me question if the new cost function works efficiently with other recourse frameworks, e.g. through SGD, or manifold based (e.g., FACE, Poyiadzi et al., 2019) or latent space methods (e.g., Pawelczyk et al., 2020).

**Summary:** This work tackles the relevant problem of learning better cost functions for recourse. However, the results are not unexpected  and it seems that for the most challenging issue of modeling mutual feature dependencies, manual specification is still necessary. This makes it hard for me to see why this approach would be substantially better and less time-consuming than directly specifying constrains in an optimization-based recourse framework, which on the other hand come with guarantees that the constraints will be obeyed. In conclusion, while this work is technically sound and the write-up good to follow, I remain a bit doubtful that the proposed method has substantial advantages in practice.


----------------
**References**

Ustun, B., Spangher, A., & Liu, Y. (2019, January). Actionable recourse in linear classification. In Proceedings of the conference on fairness, accountability, and transparency (pp. 10-19).

Amir-Hossein Karimi, Gilles Barthe, Borja Balle,and Isabel Valera. Model-agnostic counterfactual explanations for consequential decisions. In International conference on artificial intelligence and statistics ,pp.895–905.PMLR,2020.

Rafael Poyiadzi, Kacper Sokol, Raul Santos-Rodriguez, Tijl De Bie, and Peter Flach. 2020. FACE: Feasible and Actionable

Martin Pawelczyk, Klaus Broelemann, and Gjergji Kasneci. 2020. Learning model-agnostic counterfactual explanations for tabular data. In Proceedings of The Web Conference. Association for Computing Machinery, New York, NY, USA.

**Questions:**

* Can the authors specify more details about the independence test used?
* What are the user study scores without the undecisive cases?
* Why doesn't the LLM seem to have implicit knowledge about feature dependencies in Table 1? Can the authors think of an example where the model uses its background knowledge to reason about dependencies?
* I would appreciate additional discussion on how the learned cost function can be integrated in recourse frameworks

---

> ### Author Response · Authors · 2024-11-21
> **Author(s) Response**
>
> We thank the reviewer for noting our work is contributing to an important goal under-explored in the literature, and for your suggestions which have improved the paper, we now respond to all your questions and concerns.
>
> ***
>
> **Why not use Cohan’s kappa:** Thank you for the suggestion, but Cohen’s Kappa would not work here, it is used for assessing agreement between 2 raters. Here we have 30 Human raters (and “30 LLM raters”) and are comparing two groups representing two distributions, so the Chi-Squared test of independence was an appropriate choice.
>
> ***
>
> **What exact test was used and how to interpret p values?**  We used the Chi-square test of independence (i.e., scipy.stats.chi2_contingency), which is used to compare 2 or more groups. Apologies for the lack of clarity, we are hoping that the p value will be > 0.05, because we want to demonstrate that the distributions of LLM and human are not significantly different.  We state this clearly in Figure 2’s caption now. (ln 264)
>
> ***
>
> **How would the results change if scores where only the accepted pairs by the LLM are used were obtained?** If we remove all instances where the LLM was uncertain we would lose 6 questions where the LLM was 100% uncertain. The remaining 12 would have some with uncertain LLM responses. Of these, 8/12 have statistically similar distributions, which is proportionally similar to the 13/18 reported in the paper.
>
> ***
>
> **Selection of examples:** The samples were selected according to their mean values in the dataset and perturbed by standard deviations. This guaranteed they would be reasonable feature values. The questions were purposefully chosen to be very difficult, which is evidenced by the uncertainty shared in the LLM and humans (49% uncertain ln 270). To illustrate this we added additional analysis in Appendix F showing the uncertainty in users and LLM were both equally high and strongly correlated between questions. This shows the questions were not obvious. In total we are evaluating three desiderata across 18 questions, which is not low by the standards of recent recourse publications which sometimes just evaluate as low as six questions with humans [1]. We added the uncertainty results in Appendix F and mention them in Section 4.1
>
> ***
>
> **What is the advantage of using the LLM:** Our LLM method is the only method which can model feature dependencies naturally, this is a significant advantage on all prior work. If we compare it to other methods like Ustun et al. (2019) they will fail on this by definition, so it would be a trivial comparison. Note we already showed this in the human study when the LLM correlated with human judgement on Desideratum 3, and we have added additional computational test in Section 4.3 to show this now more explicitly.
>
>  ***
>
> **The LLM seems to fail to inherently model dependencies.** Please see above response. The LLM does naturally model these dependencies, the original results in Table 1 were us purposefully choosing dependencies which were not modelled to show we could fine-tune them into the cost function.  We have added computational dependency tests in Section 4.3, and shown how dependencies can be fine-tuned in Section 4.4. with our custom prompting scheme.
>
>  ***
>
> **Apply the cost function in popular recourse frameworks:** We have replaced our L1 baseline in Section 4.3 with (1) an SGD-based method by Watchter et al. [3] and (2) a data driven method by Smyth & Keane [2] to test our models, both highly cited and contrasting methods.
>
>  ***
>
> **Can the authors specify more details about the independence test used?** We used the Chi-square test of independence for each question to determine if there is a significant difference in the proportion of each choice between the two groups (30 LLM responses and 30 human responses in 18 questions, 540 per group).
>
>  ***
>
> **What are the user study scores without the undecisive cases?** See above responses.
>
>  ***
>
> **Why doesn't the LLM seem to have implicit knowledge about feature dependencies in Table 1**
>
> We purposefully chose dependencies which did not perform well to illustrate how fine-tuning can help in Table 1. Please note the human studied showed how the LLM picked up on natural dependencies in Desideratum 3 (6/6 had statistically similar distributions or same mode). We have added Section 4.3 which is a new dependency experiment to show where the model can reason about dependencies.
>
>  ***
>
> **I would appreciate additional discussion on how the learned cost function can be integrated in recourse frameworks**
>
> We have shown an example in our case study now for two published methods [2,3], thanks for the suggestion.
>
> ***
>
> [1] Kenny E, Huang W. The Utility of “Even if” semifactual explanation to optimise positive outcomes
>
> [2] Keane MT, Smyth B. Good counterfactuals and where to find them.
>
> [3] Wachter S, Mittelstadt B, Russell C. Counterfactual explanations without opening the black box

---

> ### Comment · Reviewer_8sk6 · 2024-11-22
> **Response**
>
> Thank you for your reply which has clarified the following of my points:
> * **Cohen's Kappa**: Yes, the classical kappa only works for 2 raters, but there are extensions to  multiple dimensions, e.g., https://en.wikipedia.org/wiki/Fleiss%27_kappa (but that is a minor point for me).
> * **Chi2-test** I understand the type of test that was performed now which should be fine. Maybe mention more explicitly that you use the independence test not for independence testing but for testing a if two samples are from the same distribution. That was what confused me, when reading the draft.
> * **Selection of non-obvious examples.** Thanks for providing the plot in Appendix F which shows that the examples selected were non-trivial.
> * **Modeling dependencies.** I am still a bit confuesed about the selection of dependencies for which some of them are picked up and some are not, but I like the new experiment in Section 4.3. I definitely strengthens this aspect of the work. Some discussion about how the dependencies in Table 1 were chosen and which dependencies the LLM usually does not pick up would be appreciated.
>
> * **Baselines.** The results in the Case study show that the new cost function can be used with some recourse techniques (of course simple ones), but this is an improvement.
>
> My following concerns remain:
>
> * **LLM Uncertainty rate in the study is too high.** I agree with the other reviewers, that the high uncertainty rate is problematic. As the authors point out it reduces the small number of 18 examples further to 12 samples for which there is an actual signal.
>
> I thank the authors for their thorough rebuttal, which has addressed most of my concerns. I have updated my score from 5->6.

---

> > ### Author Response · Authors · 2024-11-27
> > **Author(s) second response**
> >
> > Many thanks for reconsidering your score, here we address your remaining concerns.
> >
> > ***
> >
> > *Chi2-test... Maybe mention more explicitly that you use the independence test not for independence testing but for testing a if two samples are from the same distribution.*
> >
> > We have updated Figure 2's caption to mention this explicity.
> >
> > ***
> >
> > *Modeling dependencies. I am still a bit confused about the selection of dependencies for which some of them are picked up and some are not. Some discussion about how the dependencies in Table 1 were chosen and which dependencies the LLM usually does not pick up would be appreciated.*
> >
> > To answer your first question, we chose ground truth dependencies for Section 4.3 using Claude Sonnet 3.5 by prompting it 10 times and took the most common three (see Appendix G). These were then evaluated in our pipeline, and found 6/9 were modelled with the “correct” polarity in our MLP cost functions.
> >
> > For your second point, the dependencies in (what was) Table 1 were chosen to be the worst performing ones in our previous test, we want to see if we can fine-tune the LLM prompt, and hence the final cost function, to model these “correctly”. The dependencies we fine-tuned are in Appendix H.
> >
> > Please note the reason some “failed” is because some of these dependencies listed by Claude were somewhat subjective, such as e.g. “It is hard to get divorced if you are older”. So, to make this evaluation more convincing, we added an additional test which accounts for this by only considering known scientific causal dependencies. Specifically, we have added additional tests to Section 4.3 on synthetic personal medical data, these show that the LLM successfully models the cost of known scientific causal dependencies with > 90% accuracy (note, we used synthetic medical data to be as sure as possible no similar counterfactual data is in the pre-training LLM data).
> >
> > We have re-written Section 4.3 to add some more discussion about this as you ask (red highlighted text).
> >
> > ***
> >
> > *Baselines. The results in the Case study show that the new cost function can be used with some recourse techniques (of course simple ones), but this is an improvement.*
> >
> > Thank you, we have now also added the two other datasets to the case study in Appendix J. We focused on Adult Census initially because it has the most objectively inactionable features (e.g., native country U.S.).
> >
> > ***
> >
> > *LLM Uncertainty rate in the study is too high. I agree with the other reviewers, that the high uncertainty rate is problematic. As the authors point out it reduces the small number of 18 examples further to 12 samples for which there is an actual signal.*
> >
> > We appreciate your concern (and that of other reviewers), and have experimented with new prompting approaches. We updated our standard prompt to include a high-level overview of the Desiderata in Section 2, which generalizes across all datasets (see our new general response to all reviewers or Appendix D, standard prompt, red highlighted text). **This has brought the LLM uncertainty from 49% → 13.8% on average in the human study (with NO questions being 100% random)**. The alignment results are now even better (17/18 have *p* > 0.05 or the same mode), here are the full results for your convenience. We hope this alleviates your remaining concern. We have updated the paper and more details about this in Appendix F and Table 2.
> >
> > | Question | P-value | Same Most Common Response | LLM Uncertainty (%) |
> > |----------|---------|----------------------------|---------------------|
> > | 1        | 0.27    | True                       | 0.00                |
> > | 2        | 0.00    | True                       | 0.00                |
> > | 3        | 0.26    | True                       | 7.14                |
> > | 4        | 0.18    | False                      | 42.86               |
> > | 5        | 1.00    | True                       | 28.57               |
> > | 6        | 1.00    | True                       | 46.43               |
> > | 7        | 1.00    | True                       | 0.00                |
> > | 8        | 0.53    | True                       | 0.00                |
> > | 9        | 0.35    | True                       | 0.00                |
> > | 10       | 0.53    | True                       | 3.57                |
> > | 11       | 0.47    | True                       | 0.00                |
> > | 12       | 0.59    | False                      | 71.43               |
> > | 13       | 0.00    | False                      | 0.00                |
> > | 14       | 0.03    | True                       | 0.00                |
> > | 15       | 0.06    | True                       | 0.00                |
> > | 16       | 0.56    | True                       | 32.14               |
> > | 17       | 0.12    | True                       | 0.00                |
> > | 18       | 0.24    | True                       | 17.86               |
> >
> >
> > ***
> >
> > We hope the reviewer’s remaining concerns are now fully addressed.
> >
> > Best wishes
> >
> > The Author(s)

---

> > > ### Author Response · Authors · 2024-12-01
> > >
> > > Dear Reviewer 8sk6,
> > >
> > > We appreciate you are busy, but as we are approaching the final day of the discussion period, we hope we have adequately addressed your remaining concerns. Please let us know if there are any further clarifications or additional points you would like us to discuss, or if we have adequately addressed your remaining concerns.
> > >
> > > Best wishes,
> > >
> > > The Authors

---

### Official Review · Reviewer_umaS · 2024-11-02

**Soundness:** 2
**Presentation:** 2
**Contribution:** 2
**Rating:** 3
**Confidence:** 4

**Summary:**

The authors study empirically the problem of learning a cost function for algorithmic recourse by replicating a Bradley-Terry model from LLM-generated data. The authors provide a framework to generate a suitable dataset, by asking an LLM to judge pairs of recourse options. Lastly, the authors evaluate their approach by training and MLP and a tree-based model to replicate the LLM-generated comparison data on three different datasets taken from the literature.

**Strengths:**

The problem studied by the authors is interesting and timely. The lack of suitable cost functions for recourse is a long-standing problem in the community, and the lack of reasonable data limits immensely the evaluation of recourse techniques. Exploiting LLMs to generate recourse-specific data is an interesting and novel application.

**Weaknesses:**

While the topic is interesting, the main issue of the present work lies in the not convincing experimental evaluation. Given that learning the cost function using a Bradley-Terry model is standard, and given that there were examples of (I believe) more realistic interactive approaches where they propose to query directly the users to elicit their cost function [1,2], I would like the evaluation of the LLM contribution to be rock solid.

**[Human-LLM Comparison]** The empirical comparison the authors presented in Section 4.1 is not enough to claim LLMs can decide as humans in the context of recourse. The analysis has mainly two issues:
- The authors ask only 6 questions for each dataset which is too few compared to the potential number of pairs $(x_i, x_i’)$ available. The number of participants is limited and might not be representative of the population (e.g., all of them are AI researchers). Lastly, the six questions might have been unintentionally cherry-picked to show this correlation.
- The LLM was unsure 49% of the time, meaning that of those 18 questions, 9 were randomly answered. The authors claimed it is fine since humans randomly select if they are unsure, however, the authors must prove this has been the case for their experiments (e.g., 49% of the answers are also given randomly by the users).

I would suggest the authors perform a more thorough experimental analysis by increasing their sample size and the number of comparisons and evaluating how many times users answer randomly to their questions. Right now, the claim that LLMs can generate the same answers given to users is not convincingly confirmed.

**[Cost function evaluation]** In Section 4.2, the authors evaluate two different prompts by taking a standard and custom one (with additional information about the constraints of the cost function). However, the research questions and the description of the experiment results (Table 1 and Section 4.2) are hard to understand. For example, it is not clear what ground truth the authors compare against in all the experiments (e.g., from the text, it is generated by the LLMs, but using which prompt?). Moreover, it would have been interesting to generate a ground truth _from_ user comparison which would have yielded a more attractive comparison. Please see below also a question regarding Section 4.3).

I would suggest the authors state more clearly the research question in Section 4.2, the evaluation procedure and better describe their results in Section 4.2.2, considering also the desiderata described in Section 2.

[1] Wang, Zijie J., et al. "Gam coach: Towards interactive and user-centered algorithmic recourse." Proceedings of the 2023 CHI Conference on Human Factors in Computing Systems. 2023.

[2] Esfahani, Seyedehdelaram, et al. "Preference Elicitation in Interactive and User-centered Algorithmic Recourse: an Initial Exploration." Proceedings of the 32nd ACM Conference on User Modeling, Adaptation and Personalization. 2024.

**Questions:**

- Why did you not use a specific algorithm to find recourses while you relied simply on permutations?
- What is the point of pairing the recourses in a graph (Section 3.2)?
- What is the baseline used to evaluate the approaches in Section 4.2? Do you have some ground truth evaluation/cost function?
- Why do you pick only different recourses to compare against the $L_1$ norm and the trained MLP (Section 4.3)? Are you not forcing them to have a different cost? (e.g., consider three recourses A, B and C. A and B have minimal cost for the MLP, while B and C have minimal cost for the $L_1$. By picking C and A, you might say that the MLP finds cheaper options than the $L_1$ norm).

---

> ### Author Response · Authors · 2024-11-21
> **Author(s) Response**
>
> We thank the reviewer for noting our work is interesting and timely, and for your suggestions which have improved the paper, we now respond to all your questions and concerns.
>
> ***
>
> **Make the LLM contribution rock solid:**  We regret not communicating this well. The LLM is necessary to automate the labelling of pairwise comparisons at scale and model feature dependencies automatically. No prior approach can do this. Note that our human study already demonstrated this as all six Desideratum 3 (i.e., the feature dependency questions) materials achieved either statistically similar distributions or the same modal response, but we added more computational evaluation in Section 4.3 to make this even stronger.
>
>  ***
>
> **Human-LLM Comparison Issues:** First, in total there are 16 possible feature permutations per dataset, or 48 in total across three, and an additional 2 per dataset when considering mutating the same numerical feature at different ranges as a comparison. We sampled 18, which is 33% of all possible permutations and quite reasonable. Second, our number of participants is double what is observed in some recent recourse studies [1]. Third, LLMs are shown to correlate with large user studies [2], and as our group did correlate with the LLM here, it is reasonable to infer they are generally representative of the population. The six questions were intentionally chosen to use mean values in the dataset, mutate by standard deviations, and be non-trivial, which is evidenced by the uncertainty of the humans and LLMs (see new Appendix F), our questions were all selected before we saw any results, and were not cherry picked from a larger pool.
>
>  ***
>
> **Prove that 49% of the answers are also given randomly by the users:** As an aside, this does not mean 9 questions were randomly answered, it means that 49% of the responses were random, which were spread out among questions. We have included extra results on the human responses when they gauged how “far apart” recourse pairs are (i.e., the second question in Figure 6), which represents their uncertainty. We took the mean responses across these questions and the uncertainty of the LLM which was quantified by what percentage of responses were uncertain for each question. Both were then plotted in Appendix F and Figure 7. They show a strong correlation (Person r > 0.5; p < 0.02), providing evidence that not only are close to 49% of human responses also random, but **they are equivalently random for the exact same questions**. We mention this in Section 4.1 results paragraph (ln 271).
>
>  ***
>
> **Evaluation is hard to understand, what is the ground truth, why not generate human ground truth?** The human study showed our standard prompt (i.e., B in Figure 1) did well at approximating human labels (15/18). However there was misalignments, the point of the custom prompt (and our previous computational tests - now Section 4.4) was to fine-tune the labeling process to e.g. reorder feature weightings to fix minor issues in the cost function. We have added the ground truths to the paper in Appendix H. We agree it’s important to compare to a human ground truth, but we did this in the human study, so this is not a valid concern.
>
>  ***
>
> **Redo Section 4 with the desiderata:**  Agreed, we have written Section 4.4 (Fine Tuning Experiments) to do this, and iterate the desiderata 1-4 to do this as you suggest.
>
>  ***
>
> **Why did you not use a specific algorithm to find recourses:** We have replaced this section to use two diverse, highly cited algorithms, thanks for the suggestion.
>
>  ***
>
> **What is the point of pairing the recourses in a graph (Section 3.2)?**  Having a connected graph improves the reliability of our cost estimates, because it ensures the costs of all recourses can be estimated on a common scale.
>
>  ***
>
> **What is the baseline used to evaluate the approaches in Section 4.2?**  Yes, we defined the ground truth in our code previously, but neglected to add it to the paper, apologies for this, we have added it to Appendix H.
>
>  ***
>
> *Why do you pick only different recourses to compare against the L1 norm and the trained MLP (Section 4.3)? ...By picking C and A, you might say that the MLP finds cheaper options than the L1 norm).*
>
> We have updated this experiment with 6000 comparisons between methods, all validated and compared on exactly the same data. Note we are not concerned with the actual cost in this test, only the recourse ultimately chosen and the actions it results in.
>
>  ***
>
> [1] Kenny E, Huang W. The Utility of “Even if” semifactual explanation to optimise positive outcomes. Advances in Neural Information Processing Systems. 2024 Feb 13;36.
>
> [2] De Bona FB, Dominici G, Miller T, Langheinrich M, Gjoreski M. Evaluating Explanations Through LLMs: Beyond Traditional User Studies. arXiv preprint arXiv:2410.17781. 2024 Oct 23.

---

> > ### Comment · Reviewer_umaS · 2024-11-22
> >
> > I would like to thank the reviewers for their rebuttal and revised PDF. However, after reading the other reviews and answers, I am still not convinced regarding the reliability of the conducted human-LLM evaluation study. Some general points below:
> >
> > - **[Relevance of the comparisons]** I believe the recourse options are too few to constitute a reasonable benchmark to measure human-LLMs alignment. For example, assume each dataset has 3 features and assume each of them can take only 2 values. The potential recourses I can suggest to the users are 2^3*2=16. If we compute the potential questions you can make (e.g., pairs of two recourses), we get 256. If that would be the case, you would need to sample at least ~84 questions to reach your 33%.
> > - **[LLMs correlated with users]** I disagree with the authors claim. There have been studies showing also how **LLMs do not correlate with user responses** ([1-3] and many others). I would be careful to argue LLMs match human preference in general, especially in sensitive contexts like recourse. Moreover, as the authors underline in the rebuttal, the claim is in general true for "large user study", which is not the case for this paper. Lastly, given that all participants are _"thirty AI researchers"_ (line 237), the rebuttal answer _"it is reasonable to infer they are generally representative of the population"_ is not convincing (even though authors acknowledge they have a biased sample in footnote 3).
> > - **[LLMs unsure 49% of the times]** Out of the 18 questions, almost half have random answers, since the authors replace them (line 270). Again, this makes the analysis weaker given the limited sample size.
> > - **[Appendix F]** I believe the plot in Figure 7 does not show **any** uncertainty correlation between humans and LLMs (I would have expected points to **align** on the red line), strengthening the points made in the previous two comments.
> >
> > The other reviews outlined similar concerns regarding the limited evaluation, and unclear results. Thus, I will be keeping my recommendation.
> >
> >
> > [1] Zheng et al., Judging LLM-as-a-Judge with MT-Bench and Chatbot Arena. NeurIPS (2023).
> >
> > [2] Li et al., PRD: Peer Rank and Discussion Improve Large Language Model Based Evaluations. TMLR (2024)
> >
> > [3] Boubdir et al., Elo Uncovered: Robustness and Best Practices in Language Model Evaluation. Proceedings of the Third Workshop on Natural Language Generation, Evaluation, and Metrics (GEM) (2023)

---

> ### Author Response · Authors · 2024-11-27
> **Author(s) second response (part 1)**
>
> Many thanks for your response, we have addressed the evaluation and presentation concerns of other reviewers as you mention, and now we address your remaining concerns about the human study. Please note that in order to respond adequately to your concerns, we had to split this into two parts (this is part 1).
>
> ***
>
> *[LLMs unsure 49% of the times] Out of the 18 questions, almost half have random answers, since the authors replace them (line 270). Again, this makes the analysis weaker given the limited sample size.*
>
> We appreciate your concern, and have experimented with new prompting approaches. We updated our standard prompt to include a high-level overview of the Desiderata in Section 2, which generalizes across all datasets (see above general response to all reviewers, or Appendix D, standard prompt, red highlight). **This has brought the LLM uncertainty from 49% →13.8% on average in the human study., with no questions being 100% random**. The alignment results are now even better (17/18 have *p* > 0.05 or the same mode), here are the full results for your convenience.
>
> | Question | P-value | Same Most Common Response | LLM Uncertainty (%) |
> |----------|---------|----------------------------|---------------------|
> | 1        | 0.27    | True                       | 0.00                |
> | 2        | 0.00    | True                       | 0.00                |
> | 3        | 0.26    | True                       | 7.14                |
> | 4        | 0.18    | False                      | 42.86               |
> | 5        | 1.00    | True                       | 28.57               |
> | 6        | 1.00    | True                       | 46.43               |
> | 7        | 1.00    | True                       | 0.00                |
> | 8        | 0.53    | True                       | 0.00                |
> | 9        | 0.35    | True                       | 0.00                |
> | 10       | 0.53    | True                       | 3.57                |
> | 11       | 0.47    | True                       | 0.00                |
> | 12       | 0.59    | False                      | 71.43               |
> | 13       | 0.00    | False                      | 0.00                |
> | 14       | 0.03    | True                       | 0.00                |
> | 15       | 0.06    | True                       | 0.00                |
> | 16       | 0.56    | True                       | 32.14               |
> | 17       | 0.12    | True                       | 0.00                |
> | 18       | 0.24    | True                       | 17.86               |
>
> ***
>
> *[Appendix F] I believe the plot in Figure 7 does not show any uncertainty correlation between humans and LLMs (I would have expected points to align on the red line), strengthening the points made in the previous two comments.*
>
> Given the amount of variability and noise in real-world data (and the space of possible values), it is often the case that points will not align on the red line for modest samples as we have (18 questions). Please note the objective statistical test Person *r* was 0.55 (*p* < 0.02) – [now updated to *r*=0.48 and *p*=0.04 with the new data collected from the LLM], which is considered a strong-moderate correlation. In either case, we have updated this plot now with the new data, and we feel it is a stronger visual correlation (as the objective statistical measurements again verify), one point even lies directly on the red line as the reviewer wished to see to be more convinced (although now it is perhaps less relevant since the uncertainty in the LLM has dropped so much).

---

> > ### Author Response · Authors · 2024-11-27
> > **Author(s) second response (part 2)**
> >
> > *[Relevance of the comparisons] I believe the recourse options are too few to constitute a reasonable benchmark to measure human-LLMs alignment. For example, assume each dataset has 3 features and assume each of them can take only 2 values. The potential recourses I can suggest to the users are 2^3*2=16. If we compute the potential questions you can make (e.g., pairs of two recourses), we get 256. If that would be the case, you would need to sample at least ~84 questions to reach your 33%.*
> >
> > We kindly point out that our paper focused on single feature mutations (we originally pointed this out in Section 3.1), so within that scope we did actually sample a reasonable number of permutations. Note however we did also experiment with multiple mutations, and found our method did generalize well there too (Appendix C). Apologies if that wasn’t clear initially.
> >
> > ***
> >
> > *[LLMs correlated with users] I disagree with the authors claim. There have been studies showing also how LLMs do not correlate with user responses ([1-3] and many others). I would be careful to argue LLMs match human preference in general, especially in sensitive contexts like recourse. Moreover, as the authors underline in the rebuttal, the claim is in general true for "large user study", which is not the case for this paper. Lastly, given that all participants are "thirty AI researchers" (line 237), the rebuttal answer "it is reasonable to infer they are generally representative of the population" is not convincing (even though authors acknowledge they have a biased sample in footnote 3).*
> >
> > Thank you for acknowledging our scientific integrity. We are well aware of the mixed results comparing LLMs and users in testing (we talk about it in the last paragraph of our Related Work), so we were careful to run a human study before making *any* claims about alignment in judgement of cost, so we have always been aligned on this with the reviewer. As an aside, the papers the reviewer referenced all had their original versions published in June/July/Nov 2023, and much progress has been made since then (see Oct 2024 [2]).
> >
> > We pointed out “large” user studies before because it demonstrates higher probability the conclusion is true, but this does not mean that smaller user studies cannot show the same thing, that is part of what our statistical tests are for. For instance, the probability that 17/18 questions are aligned (as they are) in our user study, by chance, is highly unlikely. So again, even though the sample is relatively modest in size, the statistical significance of the results are so strong, that it mitigates this concern.
> >
> > Every user study has some sampling bias (e.g., only U.S. participants, only Amazon Turk/Prolific Users which by definition biases towards users with internet access and some tech ability). While all participants were data scientists, there was a balance of male/female, non-native/native English speakers, nationality, and age, we disclosed all this information in the paper so readers may interpret it as they wish to contextualize it with other research results, as we did in all our user testing previously.
> >
> > ***
> >
> > *The other reviews outlined similar concerns regarding the limited evaluation, and unclear results. Thus, I will be keeping my recommendation.*
> >
> > Our evaluation is over 60% of the paper, covers human studies, dependency evaluation (on real and synthetic data now), fine-tuning experiments across all desiderata, fidelity between the cost function/LLM, and case-studies across all datasets. In addition, we have re-written the unclear parts with the suggestions from all four reviewers. We hope the reviewer is slightly more convinced now. If not, let us know, we are happy to clarify more.
> >
> > ***
> >
> > [1] Hopkins, A.K. and Renda, A., 2023, October. Can llms generate random numbers? evaluating llm sampling in controlled domains. Sampling and Optimization in Discrete Space (SODS) ICML 2023 Workshop.
> >
> > [2] De Bona, F.B., Dominici, G., Miller, T., Langheinrich, M. and Gjoreski, M., Evaluating Explanations Through LLMs: Beyond Traditional User Studies. In GenAI for Health: Potential, Trust and Policy Compliance. (2024)

---

> > > ### Comment · Reviewer_umaS · 2024-11-28
> > >
> > > **[Correlation between LLMs and users & Appendix F]** Given the few data points coming from the user's study, I would not trust any statistical results coming from such an analysis. Moreover, even if the reported correlation is high, the data might exhibit very different patterns (e.g., see Anscombe's quartet examples). Thus, I am still very sceptical about the relevance of Figure 7 in the Appendix. Moreover, by comparing the correlation graphs in Appendix F, before and after the current revision, it occurs to me that it seems the only thing that changed is the human uncertainty (e.g., the points are "squashed" at the top of the graph). Since the authors should have only changed the LLM's uncertainty data, how is it possible such behaviour?
> > >
> > > Lastly, I do not doubt that LLMs might exhibit some form of alignment, but there must be reasonable experimental evidence, e.g., in [2] the dataset consists of 731 participants, more than 20x the ones used by this study and the authors in [2] do claim the LLMs might align well just because it might have seen the data during training.
> > >
> > > The last point is also a critique reviewer xrVn raised, but I do not think the additional experiment in Appendix K satisfactorily addresses it. Indeed, if the generative process is based on _[...] scientifically known causal dependencies in medicine [...]_ I am very sure the LLMs did see these dependencies during training, and it should be more interesting to study _why_ is not able to model such dependencies (unless it is explicitly prompted for them).
> > >
> > > **[Single or multiple feature mutations]** Thank you for the clarification. I am still not sure that by considering a single feature mutation we can derive any conclusion related to LLMs mimicking human response in recourse. Recourses can be composed of multiple feature changes, and the cost function might exhibit non-linear relationships between components (e.g., changing A or B in isolation is hard, but A and B together are very cheap). Thus, it feels very synthetic considering only single mutations. In Appendix C, it does seem to me that you consider how much the MLP and tree-based model recover the LLM judgments over multiple feature permutations, but such an experiment does not tell anything about human-LLM alignment over multiple feature recourses.
> > >
> > > I thank the authors for the exchange and the various answers, but my main issue with this work remains the relatively small sample size of the human judgments dataset. Unfortunately, I believe this issue hinders most of the results and statistical analysis, and I do not think it can be improved during the rebuttal period. For example, besides the uncertainty quantification (Appendix F) issue, as far as I know, the Chi-square test is reliable only with _large_ sample sizes (and this complements a question of the reviewer 8sk6). I do believe the problem and the idea are interesting, but I stand by my previous assessment.
> > >
> > > To strengthen the results, I would imagine doing the following (it is just a rough idea, so it might have other issues):
> > >   1. Given a dataset (e.g., Adult), compute a set of potential recourse options for given users by using any recourse methods (e.g., Wachter et al.,). Thus, we can focus on predicting the cost for "recourse" options and not for synthetic random permuted examples.
> > >   2. Perform a _large_ human study by making participants rate these recourse options in a pairwise fashion.
> > > 3. Once we have such data, execute again the analysis with the LLMs to understand the efficacy of the various prompt strategies.
> > > 4. If you repeat Steps 1 and 2 with another recourse method, you could evaluate your trained cost functions (MLP and tree-based) on potentially different recourses looking at OOD generalization, since the second recourse method might provide solutions qualitatively different from the first one.

---

> > > > ### Author Response · Authors · 2024-12-01
> > > > **Author(s) third response (part 1)**
> > > >
> > > > ## Summary and preamble
> > > > We thank the reviewer for their continued engagement, and we truly appreciate the chance to address their last main concern of our human study sample size (and the other smaller points).
> > > >
> > > > First however, we summarize the reviewer’s main concerns to date and how we addressed them:
> > > >
> > > > 1.	The reviewer was extremely concerned about the LLM being uncertain 49% of the time in the human study, **we addressed this and brought it down to 13.8% (a significant improvement).**
> > > > 2.	The reviewer wanted the contribution of the LLM to be “rock solid” – We clarified this by showing how only our LLM-based method can label feature dependencies with 90% + accuracy and even added additional experiments to strengthen this.
> > > > 3.	The reviewer was worried about the unclear motivation for Section 4’s experiments, we rewrote it using the suggestions of all reviewers (most of which are now satisfied).
> > > > 4.	The reviewer was worried we didn’t evaluate on real recourse algorithms, we fixed this and evaluated on two diverse highly cited ones across all datasets.
> > > > 5.	The reviewer had three additional questions about recourse pairings, ground truths, and the case study. We trust that our clarifications on these points addressed the reviewer’s concerns, as no further clarifications were requested
> > > >
> > > > If our following rebuttal does not fully address the reviewer’s remaining concerns, we hope the extensive revisions and rigorous responses we have already provided demonstrate the value of our work and warrant a positive reconsideration of the score.
> > > >
> > > > ***
> > > >
> > > > ## Main concern: sample size in human study
> > > > The reviewer’s remaining concerns (e.g., Fig. 7 and human alignment) primarily stem from their perception that our human study (N=30) involves an insufficient sample size (*“…my main issue with this work remains the relatively small sample size of the human judgments…”*). However, they do acknowledge the significance of our results (*“…the reported correlation is high…”*).
> > > >
> > > > While we acknowledge that larger sample sizes can sometimes provide more confidence, we must balance this with statistical justification. Many papers are rejected for oversampling because it can be used to *p*-hack [1] (indeed, we were almost rejected before for having “too many” users in a study).
> > > >
> > > > If we do a statistical power analysis to determine the sample required for a chi-squared test of independence in a 2x2 contingency table (as ours is). Assuming a large effect -- Cramer's V=0.5 (because as the reviewer rightly notes we should be very skeptical that LLMs and humans correlate), an alpha $\alpha$=0.05, and a power $(1-\beta)$=0.8 (i.e., their standard values), the analysis indicates that a total sample size of 32 participants would be sufficient to detect the effect. Our actual sample size is 30, which provides a reasonable level of power, minimizing the risk of Type II error. Given the strength and significance of our results, increasing the sample size further would not meaningfully impact the conclusions.
> > > >
> > > > Furthermore, two impactful XAI papers (i.e., [2,3] -- cited together nearly 50,000 times) conducted human studies with 30 and 27 participants, demonstrating that even modestly sized studies can yield influential findings when thoughtfully designed and rigorously analyzed.
> > > >
> > > > We believe our study provides a strong foundation for future work to build upon. Larger-scale studies could explore broader applications or refine specific insights, but increasing the sample size in our case would not substantively alter our conclusions.

---

> > > > > ### Author Response · Authors · 2024-12-01
> > > > > **Author(s) third response (part 2)**
> > > > >
> > > > > ## Other Secondary Concerns:
> > > > >
> > > > > *..the Chi-square test is reliable only with large sample sizes*
> > > > >
> > > > > The Chi-squared test of independence has no hard assumptions about the sample size, it is just generally suggested that the expected values in cells are > 5 for 80% of the cells [4], for us, 87% of the cells have an expected value > 5.
> > > > >
> > > > > ***
> > > > >
> > > > > *… the authors in […] do claim the LLMs might align well just because it might have seen the data during training.*
> > > > >
> > > > > We have shown in our new experiments (thanks to Reviewer xrVn’s suggestions) that the LLM can reason about datasets with 90% + accuracy which do not exist in its training data (we addressed your sample size concern in part 1).
> > > > >
> > > > > ***
> > > > >
> > > > > *The last point …  reviewer xrVn raised… if the generative process is based on [...] scientifically known causal dependencies in medicine [...] I am very sure the LLMs did see these dependencies during training, and it should be more interesting to study why is not able to model such dependencies (unless it is explicitly prompted for them).*
> > > > >
> > > > > We agree it would be interesting to study why you need the desiderata and chain-of-thought for accurate dependency labeling, but that is a question for the mechanistic interpretability community, not the recourse one.
> > > > >
> > > > > Regarding Reviewer xrVn, our interpretation was they seemed concerned about counterfactual data existing for popular datasets online, not general learned causal dependencies, which we are, in actual fact, hoping to purposefully exploit. So, we addressed that specifically by making a synthetic dataset.
> > > > >
> > > > > ***
> > > > >
> > > > > *Recourses can be composed of multiple feature changes… Appendix C… does not tell anything about human-LLM alignment over multiple feature recourses.*
> > > > >
> > > > > 16.6% of our user study questions actually did do multiple feature mutations (see supplement), all of which showed statistical agreement with humans. This was to reflect our permutation function (Appendix B), which was 80% focused on single mutations, and 20% on multiple.
> > > > >
> > > > > Importantly, we primarily considered single feature mutations because our finetuning experiments require this. E.g., we can’t mutate two features when trying to evaluate Desideratum 1/2. Most researchers and psychological evidence point towards recourse needing to be sparse (1-2 features max [5, 6]), so it makes sense to focus on this initially. Again, we also have computational evidence our method works fine for multiple mutations (Appendix C).
> > > > >
> > > > > ***
> > > > >
> > > > > *To strengthen the results, I would imagine doing the following…*
> > > > >
> > > > > Thanks for this, but if we did as you suggest, we would need to cherry pick examples to focus on a diverse set of features, as certain recourse methods will be biased towards certain suggestions (see our case studies, Table 2 in particular), and the reviewer was rightfully against such cherry picking previously. Also, defining a “large” user study is subjective, no matter how many we sample, the criticism of “this is too small” is always possible, which is why we need power analysis. Lastly, by perturbing many prompt variations we run the risk of overfitting to our 3 datasets. In contrast, it is more convincing that we just use the high-level desiderata other researchers defined previously (which we did, and works well already).
> > > > >
> > > > > ***
> > > > >
> > > > > Figure 7 Data: We mixed up the axis labelling, thanks for pointing that out.
> > > > >
> > > > > ***
> > > > >
> > > > > We sincerely thank the reviewer once again, we have worked diligently to address all concerns raised. While we recognize there is always room for further exploration, we are confident that our study offers valuable insights and a strong foundation for future research in this area.
> > > > >
> > > > > Thank you for considering our rebuttal.
> > > > >
> > > > >
> > > > > ***
> > > > >
> > > > > [1] Faber J, Fonseca LM. How sample size influences research outcomes. Dental Press J Orthod. 2014 Jul-Aug;19(4):27-9. doi: 10.1590/2176-9451.19.4.027-029.ebo. PMID: 25279518; PMCID: PMC4296634.
> > > > >
> > > > > [2] Ribeiro, M.T., Singh, S. and Guestrin, C., 2016, August. " Why should i trust you?" Explaining the predictions of any classifier. In Proceedings of the 22nd ACM SIGKDD international conference on knowledge discovery and data mining (pp. 1135-1144).
> > > > >
> > > > > [3] Scott, M. and Su-In, L., 2017. A unified approach to interpreting model predictions. Advances in neural information processing systems, 30, pp.4765-4774.
> > > > >
> > > > > [4] https://libguides.library.kent.edu/spss/chisquare
> > > > >
> > > > > [5] Keane, M.T. and Smyth, B., 2020. Good counterfactuals and where to find them
> > > > >
> > > > > [6] Medin, D.L., Wattenmaker, W.D. and Hampson, S.E.: Family resemblance, conceptual cohesiveness, and category construction. Cognitive psychology, 19(2), pp.242-279 (1987).

---

### Official Review · Reviewer_N4gt · 2024-11-03

**Soundness:** 2
**Presentation:** 4
**Contribution:** 4
**Rating:** 8
**Confidence:** 4

**Summary:**

The paper presents a method to learn cost-functions for recourse using LLMs. It uses LLMs to label pairwise comparisons comparing the ease-of-modification costs for recourse, and then fits models on this labelled data. A user study is conducted to show that LLM labels match human labels. In order to improve the LLM labelling capability, information about the desired cost functions is included in the LLM prompts. Spearman's correlation coefficient is used to determine whether the LLM output labels match this apriori information from the prompt, and additional model accuracy is reported upon being fit to the LLM label data.

**Strengths:**

This is a good paper and I would like to see it accepted. It presents a refreshingly original method to tackle a real-world problem that has been persistent in the literature. The use of LLMs to learn recourse costs is promising.

**Weaknesses:**

My primary concerns with this paper lie in the evaluation strategies employed. In the first experiment, the LLM is reportedly unsure of the answer 49% of the time, which suggests that more creative approaches may be needed to reliably elicit cost judgments from LLMs. I would recommend that the authors explicitly address this high level of uncertainty, including a discussion of its impact on the overall feasibility and reliability of their proposed method.

In the following experiments, the conclusions are challenging to interpret. Including information about the desired cost behavior in the prompt improves Spearman correlation coefficients, and models trained on these labels show higher accuracy. However, the use of a “custom prompt” seems unrealistic as it assumes an oracle can encode information about recourse costs apriori. The paper’s stated goal is to learn cost functions for recourse, so beginning with existing cost functions only to later recover them may not be particularly meaningful. While including the answer in an LLM prompt naturally brings its output closer to the target values, I don’t see the value in aiming to recover an answer that was effectively already embedded in the prompt. I would suggest the authors clarify how these results add practical value, beyond simply reproducing known information within the prompt. Demonstrating how their method might perform in more realistic scenarios would strengthen this section.

Similarly, the accuracy comparison in Table 1 - showing how well LLM responses are modeled by either a tree-based model or an MLP - is difficult to interpret in terms of the paper’s objective. I recommend the authors expand on the significance of these comparisons, particularly in light of their goal to develop effective recourse cost functions. Further clarification on the insights this offers for their method’s performance would make this section more impactful. It seems to me that the choice of model here should be agnostic to the objective of the paper. The experiments show that pairwise comparisons can be learned with reasonable accuracy, which I think is enough to justify the paper’s merit, but the relative accuracy comparisons confused me - perhaps I missed the underlying point?

**Questions:**

I'm unable to understand the comparisons between the standard and custom prompts, and the bearing they have on the use of LLMs to learn recourse cost functions. Perhaps a search over LLM prompts (without including the desired answer directly) to best match human intuition would be more valuable. Similarly, the accuracy comparison reported in table 1 needs better justification: what does the comparison indicate - are we just checking to ensure that LLM output labels can indeed be learned by the models?

---

> ### Author Response · Authors · 2024-11-21
> **Author(s) Response**
>
> We thank the reviewer for vouching for the acceptance of our work, and for your suggestions which have improved the paper, we now respond to all your questions and concerns.
>
> ***
>
> **High uncertainty in LLM Human Study Responses:** Thanks, we purposefully chose very difficult questions for the user study which were hard to evaluate, which is why the uncertainty is high. In our main computational experiments (now Section 4.4), the LLM is unsure about the answer (i.e., gives equal cost) 15%, 23%, and 20%, of the time (using the standard prompting scheme) on adult, HELOC, and German Credit, respectively, which is much lower than the user study uncertainty, illustrating this point for us. As an aside, this dropped further to 10%, 5%, and 11% when fine-tuning using the custom prompt schemes. We explicitly mention this in Footnote 4 now, and add additional analyis about the uncertainity between LLM and humans in Appendix F.
>
> ***
>
> *...the use of a “custom prompt” seems unrealistic as it assumes an oracle can encode information about recourse costs apriori... beginning with existing cost functions only to later recover them may not be particularly meaningful...*
>
> Allow us to clarify, there has been a large misunderstanding. The purpose of the custom prompt in Section 4.2 is not to hard-code the cost function (this would be pointless as you point out). It is to show how the original cost function could be “fine-tuned” with small prompt additions. For example, to reorder feature importance, add/exaggerate a single dependency, or manipulate a continuous feature’s cost spectrum, or fairness attributes (how you define them).
>
> **Revision:** We have re-written some of Section 4 (the evaluation) to be clearer about all this by e.g. titling Section 4.4 “Fine Tuning Experiments”. Moreover, we added additional experiments showing how the LLM naturally picks up feature dependencies, which is a significant advancement over prior work.
>
>  ***
>
> *Similarly, the accuracy comparison in Table 1 - showing how well LLM responses are modeled by either a tree-based model or an MLP - is difficult to interpret in terms of the paper’s objective... the choice of model here should be agnostic to the objective of the paper. The experiments show that pairwise comparisons can be learned with reasonable accuracy, which I think is enough to justify the paper’s merit, but the relative accuracy comparisons confused me - perhaps I missed the underlying point?*
>
> We considered two model classes because they have distinct practical advantages. Only MLPs are differentiable (which is needed by many recourse methods), and the tree models have more ability to be transparent, which is important in domains such as medicine or finance where interpretability is a strong requirement, so the comparison is quite important. We are using accuracy to verify how well they are imitating the original LLM labeling process, which is important as we are essentially trying to distill the LLM’s knowledge into smaller cost functions.
>
> **Revision:** We explain why we compared MLP and tree model accuracy better (ln 303 and Section 4.5).
>
>  ***
>
> *I'm unable to understand the comparisons between the standard and custom prompts, and the bearing they have on the use of LLMs to learn recourse cost functions. Perhaps a search over LLM prompts (without including the desired answer directly) to best match human intuition would be more valuable. Similarly, the accuracy comparison reported in table 1 needs better justification: what does the comparison indicate - are we just checking to ensure that LLM output labels can indeed be learned by the models?*
>
> Apologies for the confusion, the standard prompt was used in the human study, and this showed strong results (15/18 mostly aligned). However, there was room for improvement, so the custom prompt is there to fine-tune the LLM responses to match preferences in a specific domain (with minimal instructions). We did not see the need to iterate more LLM prompt variations as we risk overfitting to our human study results, and the standard prompt (i.e., no bias injected in the prompt from us) already did quite well, and we anticipate this will just get better with later LLM models. See above response for accuracy concerns in Table 1.
>
> **Revision:** We titled Section 4.4. “Fine Tuning Experiments” and made the purpose of the custom prompt clear in Section 4.2 (ln 296).

---

> ### Comment · Reviewer_N4gt · 2024-11-24
>
> High uncertainty in LLM Human Study Responses - Upon clarification, it is interesting that the human uncertainties align with the LLM uncertainties. I also see more clearly the intentions behind the use of the MLP and the decision trees.
>
> My question about the unrealistic custom prompt remains. There seems to be a misunderstanding here, as per the authors, but I am still unsure my concerns have been addressed. I think I follow this statement: *The purpose of the custom prompt in Section 4.2 is not to hard-code the cost function (this would be pointless as you point out).*. However the following, to me, seems to contradict it: *the original cost function could be “fine-tuned” with small prompt additions. For example, to reorder feature importance, add/exaggerate a single dependency, or manipulate a continuous feature’s cost spectrum, or fairness attributes (how you define them).* Isn't this description doing exactly what my initial question was about - inserting the "answer" in the prompt, and having the LLM recover it in the form of pairwise comparisons used to fit the cost function?
>
> To me, the "fine-tuning" of the prompt by modifying it apriori with properties desired in the cost function, and then observing that the cost function is better attuned to those properties, seems counterproductive. This may not be "hard-coding", but is still essentially leaking information about the cost function to the LLM via the prompt, when the point of the paper (as I understand it) is to elicit the cost function from the internal knowledge of the LLM. If the authors could help clarify if/what i am misunderstanding, I would be like to update my score. Like I mentioned, I think this is a valuable contribution and a unique approach that needs to be shared with the community, but this particular experiment seems to indicate that perhaps LLMs are not indeed a reliable way to learn high-performing cost functions for recourse. If they were, why would we need to include desirable properties about the cost function, known apriori from external domain knowledge, in the prompt?

---

> > ### Author Response · Authors · 2024-11-27
> > **Author(s) second response**
> >
> > Please accept our sincere apologies for the confusion, we understand now and appreciate your clarifications greatly. Yes, it is true that the LLM largely correlates with human intuition of cost (now 17/18 of our human study questions show positive results with our updated version of the standard prompt discussed in the general response to reviewers and AC), but it is also true that it will need to be fine-tuned sometimes, we will give you a few examples.
> >
> > Firstly, the definition of fairness in machine learning is currently domain specific [1]. It will be true that legally, in certain domains, we must have zero demographic bias when mutating features for recourse. However, it will also be conversely true, that in other domains, we must acknowledge systemic barriers that exist for certain groups, in which case we must add bias (or leave the current bias alone). We showed in our fine-tuning experiments that we can “leave alone/add/remove” similar dependencies in Desideratum 3 and 4.
> >
> > As another example, in finance, during an economic downturn, a bank may need to adjust recourse algorithms to prioritize different features for lending criteria, in such case the cost of certain features needs to be changed relative to others, which our fine-tuning experiments showed within our Desideratum 1 evaluation was possible.
> >
> > In healthcare, often patent safety may need to be prioritized over cost, so instructing the LLM in its prompt to consider specific feature weightings for cost is necessary also to help guarantee this happens, which we showed was possible in Desideratum 1 when we re-ordered the feature costs relative to each other.
> >
> > By putting these into the prompt, it gives us a ground truth for our experiments to verify if the final cost function was fine-tuned correctly.
> >
> > Thank you for the question, this is important, we have added a few sentences to the start of Section 4.4 to make this explicit and improve the clarity and presentation (highlighted in red). Again, thanks a lot for pointing this out.
> >
> > ***
> >
> > [1] Mehrabi, N., Morstatter, F., Saxena, N., Lerman, K. and Galstyan, A., 2021. A survey on bias and fairness in machine learning. ACM computing surveys (CSUR), 54(6), pp.1-35.

---

> > > ### Comment · Reviewer_N4gt · 2024-11-27
> > >
> > > This was a very useful clarification, thank you. All the concerns I had initially raised seem to now be addressed. I am updating my score accordingly, and I look forward to the community building upon this preliminary work to further "tune" cost functions in different domains.

---

### Official Review · Reviewer_xrVn · 2024-11-04

**Soundness:** 3
**Presentation:** 3
**Contribution:** 2
**Rating:** 5
**Confidence:** 4

**Summary:**

This paper proposes a novel approach for obtaining a cost function for recourse. In the first step, an LLM (GPT-4o in this study) is queried to decide which recourse actions are low-cost. In the second step, a cost function is learned based on the LLM responses. The points at which the LLM is queried are selected to obtain good coverage of possible recourse actions.

The paper considers four desirable criteria (feature cost, relative cost, dependant cost, and fair cost) along which the final cost function is evaluated.

The paper uses chain-of-thought prompting and finds that instructing the LLM with the desired cost function criteria improves the LLM responses.

The paper also compares human and LLM judgements of cost in a study with human participants, finding that there is often alignment, but not always.

**Strengths:**

The research question studied in this paper is interesting and of practical relevance.

The proposed methodology is interesting and could prove helpful in designing cost functions in practice.

The paper is well-written (apart from some minor hiccups), and the experiments are reasonably well-documented.

**Weaknesses:**

**Missing Baselines:** I find it hard to judge how "high-performing" the final cost functions actually are, and also how much the proposed approach improves upon alternative approaches, because there is only very limited comparison against alternative approaches. In Section 4.3, the paper compares the recommended recourse on Adult with the recourse proposed by the L1-norm, but this is a rather limited comparison and a weak baseline. As a matter of fact, I wonder how difficult the problem of designing a recourse functions actually is. For the small datasets that are considered in this study, it seems to me that it should be possible to design relatively good recourse functions "by hand", that is, with a limited amount of human labelling after taking into account the restrictions imposed by the four desiderata. Of course, and LLM might again help, but by how much does the LLM actually help here?

**Limited Evaluation:** While the newly proposed approach is well-described in the paper, it is much less discussed how a cost function should be evaluated. Indeed, this seems to be a fairly tricky question, and I suspect that the authors are well aware of this (For example: Who decides what are "relevant dependencies" in Desiderata 3). However, without clearly specifying how a cost function should be evaluated, the key claim of the paper that the obtained cost functions are "high-performing" remains subjective.

I do understand, of course, that there are evaluations in Table 1 and Figure 3 of the paper. However, these evaluations seem relatively ad-hoc to me and are not motivated in any earlier sections of the paper.

As a bit of a side note, if we were to specify very clearly how a cost function should be evaluated, should this not give rise to fairly explicit constraints on the cost function that could then be used as restrictions on the cost function during learning?

**Usage of highly popular datasets without discussion of data contamination:** This paper uses the Adult, German Credit and HELOC datasets. While these datasets are historically popular in the literature relevant to this paper, their usage with GPT-4o is problematic because of data contamination. Quite likely, GPT-4o has seen a lot of information about these highly popular datasets during pre-training. The Adult dataset, for example, is known to be partly memorized in GPT-4 (see https://arxiv.org/abs/2404.06209).

While the precise consequences of pre-training contamination on the results in this paper are very hard to judge, using (only) highly popular datasets in an evaluation with GPT-4o is not state-of-the-art. High-impact work on LLMs and tabular data (see literature references below) always attempts to include datasets that are either very recent (for example, derived from recent US census data) or datasets that have not been publicly released on the internet.

**Missing References (minor):** In the last two years, LLMs have been broadly applied to various tasks with tabular datasets other than recourse. Here are some examples:

Feature Engineering: https://proceedings.neurips.cc/paper_files/paper/2023/hash/8c2df4c35cdbee764ebb9e9d0acd5197-Abstract-Conference.html

Classification: https://proceedings.mlr.press/v206/hegselmann23a.html

Generation: https://arxiv.org/abs/2210.06280

None of this is cited in the paper. The authors should examine this literature and include it in their relative works section.

**Questions:**

- Why do you use the Adult dataset? There are better alternatives, see https://arxiv.org/abs/2108.04884 and https://github.com/socialfoundations/folktables

- Why do you choose to use latin in the paper? for example *"ad libitum"* I don't think that this terminology really helps. Also when you write *"To enforce Desideratum 2"* I would find it more helpful to write *"To enforce the relative cost criterion"*

---

> ### Author Response · Authors · 2024-11-21
> **Author(s) Response**
>
> We thank the reviewer for noting the practical relevance of our work and for your suggestions which have improved the paper, we now respond to all your questions and concerns.
>
> ***
>
> **Missing Baselines:** We did not compare against published baselines because our models already achieved high performance on all of the 4 desiderata (Section 2), including the one about feature dependencies. Hence, because no published baseline considers feature dependencies, it would be a trivial comparison, in which our method would win easily. We make this clearer in what is now Section 4.3.
>
>  ***
>
> **Weak Case-Study:**  Agreed. We have included two published recourse methods in Section 4.6, and shown how our cost function helps them achieve better recourse.
>
>  ***
>
> **Can you hard-code the cost function, how much does LLM help?** You suggestion to use a limited amount of labeled data and hard-coding the function may provide a good cost function, but it could still be quite costly as the problem of defining functions with expert knowledge is quite difficult, can take years, and needs to be constantly updated year-on-year (e.g., see [1]). This is likely why no one has proposed this in recourse, and since LLMs can largely automate the process, it is of enormous practical benefit.
>
>  ***
>
> **Limited Evaluation:** Please note we outlined clearly the dimensions along which a cost function should be evaluated as high-performing in Section 2 (which was defined by other researchers) and used it in all of Section 4 pages 4-9 showing positive results (over half the paper is dedicated to evaluation). In direct response to your concern about dependencies, we asked Claude Sonnet 3.5 to define a list of relevant dependencies in each dataset and evaluated this in Section 4.3. as a new addition to the paper.
>
>  ***
>
> **Experiments not motivated in any earlier sections of the paper:** We regret the motivation for these experiments were lost in the writing. Allow us to explain. The human study showed positive results, but that there are occasionally discrepancies between LLM and human judgment of cost. The purpose of the custom prompt and Table 1 and Figure 3 (now Fig 4/5) is to show how we can fine-tune it to fix this. For example, we saw the LLM judged certain feature-weightings to be different to humans in Figure 2 (left: Desideratum 1), so our custom prompt showed that this could be fine-tuned to correct it (see Figure 4 (middle) in revised paper). Similar results were shown for Desideratum 2-4. We re-wrote all of Section 4. to make this clearer by e.g. titling Section 4.4 “Fine Tuning Experiments”.
>
> ***
>
> **Can we specifiy constraints during learning?** We could, but this goes back to the problem of hand-defining models [1] discussed above. Our constraints are not so rigid, we mostly use the LLM’s natural labels and only occasionally modify it in our custom prompt to fine-tune the cost function, not define it from scratch, that would not scale well.
>
>  ***
>
> **Adult Data Contamination:** This is an interesting point. First, because the recourse points are synthetically generated by us, the LLM will not have seen them during its training.  Second, asking the LLM to compare recourse costs is a very different problem from predicting labels (i.e. the typical use of the datasets), and we wouldn’t expect memorization of the latter task to provide any benefit for the former. It is therefore unlikely that pre-training contamination is an issue. We point this out in the main paper and cite the work above (ln 214).
>
>  ***
>
> **Missing References:** Thanks for the suggestion, we have done as you ask (ln 504)
>
>  ***
>
> **Why do you use the Adult dataset?** We opted to use canonical datasets from the recourse literature to maximize readers’ intuition about the contexts and associated recourse cost considerations. Since the Folktables dataset you suggested is a superset of the data in Adult (drawn from the same underlying census information), we expect the results of our recourse experiments would have been virtually identical.
>
>  ***
>
> **Use of Latin and Phrasing Suggestions:**  We have replaced “ad libitum” with “at liberty”, and rephrased as you illustrated, thanks for the suggestion.
>
>  ***
>
> [1] Ruelle E, Hennessy D, Delaby L. Development of the Moorepark St Gilles grass growth model (MoSt GG model): A predictive model for grass growth for pasture based systems. European journal of agronomy. 2018 Sep 1;99:80-91.

---

> > ### Comment · Reviewer_xrVn · 2024-11-24
> > **Thanks for the author response**
> >
> > I thank the authors for their detailed response and the revised PDF. After reading the other reviews and the author's response, I stick to my original assessment of the work.
> >
> > Most of the reviewers seem to agree that there are problems with the evaluation in this paper, and I don't believe this is due to misunderstandings. In their response, the authors have already taken steps to improve the evaluation, specifically Table 1 and the revised Sections 4.3 and 4.6. I appreciate this. Table 1 goes into a good direction, I would like to see this evaluation for all the datasets. However, I also tend to agree with Reviewer umaS that there are still gaps in the evaluation: Figure 7 in Supplement F is not convincing at all.
> >
> > With regard to the point of data contamination, I appreciate that the authors are willing to incorporate this into their paper. To clarify the point from my review, it is not only about the fact that the raw datasets are memorized. It is also about the fact that hundreds of research papers are written about counterfactual explanations on these highly popular datasets, and these are very likely in the training data of the LLM. As such, while it is very hard to say how precisely the LLM becomes able to "naturally model dependencies", it does not seem absurd to believe that dependency modelling is much better for datasets and problems that the model has been heavily exposed to during pre-training (see, for example, https://arxiv.org/abs/2309.13638). This is while it is important to at least make some attempts to study the performance of the LLM for problems that are unlikely to be part of the pre-training data.
> >
> > Overall, I still believe that this paper proposes an interesting method that could be of interest to the community. In my view, the evaluation would need to be somewhat more rigorous for a venue like ICLR.

---

> ### Author Response · Authors · 2024-11-27
> **Author second response**
>
> We thank the reviewer for their response, and address their remaining concerns now.
>
> *Most of the reviewers seem to agree that there are problems with the evaluation in this paper, and I don't believe this is due to misunderstandings. In their response, the authors have already taken steps to improve the evaluation, specifically Table 1 and the revised Sections 4.3 and 4.6. I appreciate this. Table 1 goes into a good direction, I would like to see this evaluation for all the datasets. However, I also tend to agree with Reviewer umaS that there are still gaps in the evaluation: Figure 7 in Supplement F is not convincing at all.*
>
> Thanks for the kind words. To address your requests, we have run the case-study on all datasets now, with the extra results now in Appendix J.
>
> We have also addressed the evaluation concerns of Reviewer umasS you referenced by altering our standard prompt to include a high-level overview of the desiderata (see our general response to all reviewers above, or Appendix D, standard prompt, red highlighted text), the uncertainty in **the human study has dropped from 49% → 13.8%**. Figure 7 has changed now also in accordance with this, and we feel it now looks more convincing (although now it is perhaps less relevant since the uncertainty in the LLM has dropped so much). Table 2 and Appendix F describe all the results in more detail, again, the objective statistical measurements showed a significant correlation.
>
> ***
>
> *With regard to the point of data contamination, I appreciate that the authors are willing to incorporate this into their paper. To clarify the point from my review, it is not only about the fact that the raw datasets are memorized. It is also about the fact that hundreds of research papers are written about counterfactual explanations on these highly popular datasets, and these are very likely in the training data of the LLM. As such, while it is very hard to say how precisely the LLM becomes able to "naturally model dependencies", it does not seem absurd to believe that dependency modelling is much better for datasets and problems that the model has been heavily exposed to during pre-training (see, for example, https://arxiv.org/abs/2309.13638). This is while it is important to at least make some attempts to study the performance of the LLM for problems that are unlikely to be part of the pre-training data.*
>
> Thanks for the further clarification, we understand now. To sincerely address this, we have generated a dataset from scratch which does not exist on the internet with scientifically known causal dependencies in medicine and added this to Section 4.3 (details in Appendix K). Running our standard prompting scheme on this we found that **the LLM does model dependencies here with 90% + accuracy**. Interestingly, we found that in order to model the causal dependencies correctly, the LLM had to be prompted with (1) chain-of-thought, and (2) our high-level desiderata (detailed in the general response to all reviewers). Thank you a lot for this suggestion, we feel it has massively strengthened the paper.
>
> ***
>
> *Overall, I still believe that this paper proposes an interesting method that could be of interest to the community. In my view, the evaluation would need to be somewhat more rigorous for a venue like ICLR.*
>
> Hopefully our new evaluations are now rigorous enough for the reviewer.
>
>
> All the best,
>
> The Authors

---

> > ### Author Response · Authors · 2024-12-01
> >
> > Dear Reviewer xrVn,
> >
> > We appreciate you are busy, but as we are approaching the final day of the discussion period, we hope we have adequately addressed your remaining concerns. Please let us know if there are any further clarifications or additional points you would like us to discuss, or if we have adequately addressed your remaining concerns.
> >
> > Best wishes,
> >
> > The Authors

---

### Author Response · Authors · 2024-11-21
**Dear Reviewers & AC**

We would like to thank the reviewers for taking the time to read our paper and give constructive feedback. We have taken seriously the suggestions of all reviewers and either amended the manuscript accordingly to address the concern (**now uploaded, all important changes in red**), or discussed it here in our responses.

First, we thank the reviewers for their positive comments. Reviewer N4gt stated *“This is a good paper and I would like to see it accepted. It presents a refreshingly original method to tackle a real-world problem that has been persistent in the literature...”*, Reviewer xrVn noted *“The research question studied in this paper is interesting and of practical relevance.”*,  Reviewer umaS pointed out that the problem is *“interesting and timely… a long-standing problem in the community… [and] an interesting and novel application”*, and lastly Reviewer 8sk6 said *“I think it is important to have more realistic cost functions for recourse and the problem of obtaining them is under-explored in the literature. This work contributes to this goal.”*

We are grateful for these (and other) encouraging comments made.

The main concern reviewers had was with Section 4 (i.e., the evaluation). **There were two major misunderstandings:** (1) First, we did not properly communicate that **the LLM does naturally model dependencies**, which shows a large advantage of our approach over prior methods. (2) Second, **the point of our custom prompt was not to hard-code the cost function a-priori, but just to fine-tune the final cost function**. To address these points (and others), we have re-written Section 4 and added additional experiments. To summarize:

* **LLM Dependencies (Sec 4.3):** Due to misunderstandings, all reviewers were unconvinced by our evaluation and wanted a solid advantage of the LLM compared to prior work. To explain, no prior approaches can naturally label cost dependencies, we showed in our user study that we could do this (Fig. 2 Desideratum-3), but it could have been clearer. Hence, we have added another computational experiment in Sec 4.3 about dependencies to make this explicit, where we show the LLM does model dependencies reliably, and has clear advantages over prior approaches.

* **Custom Prompt (Sec 4.4):** All reviewers were confused what the point of our method is if we are hard coding the cost function into the prompt. This was a big misunderstanding, the custom prompt was just to fine-tune the cost function, not write it from scratch. We have re-written this in Section 4.4. and titled it “Fine-Tuning Experiments” to make explicit.

* **Baselines (Sec 4.5):** In response to Reviewer xrVn, umaS, and 8sk6 we have added two published recourse methods to run our cost function with, showing that our method makes current recourse methods better in practice.

* **Human Study (Sec 4.1):** In response to Reviewer N4gt, umaS, and 8sk6, we have conducted additional analysis (see Appendix F & Sec 4.1) to show that humans were correlated in their uncertainty with the LLM, so even though the LLM was  uncertain 49% of the time, this was paralleled in the humans **and shows even greater alignment than we previously thought.**

Overall, we feel our paper is now considerably stronger thanks to your feedback, and that this new version addresses all concerns by reviewers. We look forward to your feedback. Sorry about the misunderstandings, and thanks for your help in assisting us to communicate these results better.



Many thanks,

The Author(s)

---

> ### Author Response · Authors · 2024-11-27
> **Dear Reviewers & AC (note, this is the second revision of manuscript)**
>
> First and foremost, we want to sincerely thank the reviewers for taking the time to give their first round of responses, which have dramatically improved the paper. As before, we have taken seriously all your concerns and directly addressed them. All changes are again highlighted in red in the revised manuscript for your ease of reading.
>
> **Main Change:** To summarize the main change, taking to the challenge of most reviewers, we experimented with more creative prompting schemes to resolve the uncertainty concern in our human study (which was the main concern across reviews). Our discovery was that by adding a high-level description of our desiderata in Section 2 to the standard prompting scheme, **this had the effect of lowering the uncertainty of the LLM in our human study** (and improved its ability to reason about casual dependencies). Note this change to the standard prompt is general, it can be applied to any dataset without modification. Specifically, this is the addition we made to the standard prompt (corresponding to our 4 desiderata), which you can also see in Appendix D:
>
> > Remember the following 4 rules and use them in your decision:
> >
> > 1. Some features are naturally harder to change than others, use this logic.
> > 2. For numerical features, the difficulty of changing them can often depend on their starting values.
> > 3. Apart from the mutated features, consider the other features which are different between Alex and Jaden, and how this may affect difficulty.
> > 4. Do not ever use demographic features when considering the difficulty of mutating other features.
>
> To summarize the main changes:
> 1. In response to the concerns of **reviewers 8sk6, umaS, and xrVn**, we modified the standard prompting scheme to have a general description of the desiderata in Section 2 (see above or Appendix D). **We found this dropped the uncertainty of the LLM in the human study from 49% → 13.8%, with NO questions being 100% uncertain**. Moreover, it is now aligned with humans in 17/18 questions, higher than the 15/18 before. We have the full results in Table 2 and have updated Figure 7 and Figure 2 accordingly. Please note this is a general modification to the standard prompting scheme, it still can be applied to any dataset without alterations.
> 2. In response to **Reviewer xrVn**, we have extended the case study to consider all three datasets, results are in Appendix J.
> 3. In response to **Reviewer xrVn**, we have added another experiment on synthetic data in Section 4.3 (which does not exist anywhere on the internet, and hence not in the LLM’s pre-training data). The data shows that the LLM can model known scientific causal dependencies on unseen data with > 90% accuracy, helping to verify that the LLM’s ability to model dependencies is not limited to popular counterfactual/recourse generation datasets online.
>
> We have updated the paper accordingly. Thanks again for all your help to improve this paper, we would be very excited to share these results with the community.
>
> Best wishes,
>
> The Authors

---

### Meta-Review · Area_Chair_MwPF · 2024-12-21

**Metareview:**

The paper studies the problem of learning good cost functions for algorithmic recourse (so as to recommend actions to affected individuals).
The paper proposes using LLM-as-judge to label data and thereby train cost function models.
Experiments by the authors indicate that LLM judgements correlate well with human judgements,
and that LLMs are able to model feature dependencies in the cost function, unlike SOTA approaches to learning cost functions through constrained optimization.
Moreover, the authors also showed that the LLMs can be nudged effectively (in cases where their priors do not align with the required desiderata in a given domain)
simply through prompting with appropriate desiderata.

The reviewers agreed that the problem of learning recourse cost functions is well motivated and timely,
though there was significant disagreement about whether the paper's contributions are above the bar for publication at ICLR.
After a vigorous discussion among the reviewers after the rebuttal period, the main weakness identified in the paper
is that the claims in the paper are not adequately backed up by the experiment evidence.

1. "LLM is capable of reasoning about causal dependencies without being exposed to similar training data" (Appendix K). The authors created a synthetic dataset, but did not test whether LLM can answer the 3 known dependencies that they used out-of-the-box. So claiming that LLMs were not exposed to similar training data for their Appendix K experiment is an over-claim. And for the original experiments, there remain concerns around data contamination due to several papers describing the dependencies in those popular datasets such that they may be in the training data of LLMs. Overall, this issue is important because all of the domains that a practitioner may apply algorithmic recourse, we do not know that LLMs can be relied on to be a good judge.

2. The confusion around mislabeled axis in Figure 7 raised broader questions around empirical rigor -- on sufficient sample sizes, whether prompts were tuned on the test set so as to achieve significantly different results, etc. This discussion thread may well be just be an unfortunate misunderstanding sparked by an incorrect figure, so to give the authors the benefit of the doubt I did not place any weight on this point for the overall decision.

Overall, problem motivation is excellent; significance is good; clarity of exposition is poor (given how all of the reviewers shared common misunderstandings that the authors painstakingly tried to resolve during the rebuttal); soundness is below the bar (given the highly sensitive topic of algorithmic recourse, arguments that LLMs match human judgements must be done in a bulletproof evaluation).

**Additional Comments On Reviewer Discussion:**

The authors substantially improved the manuscript during the rebuttal period, thanks to the constructive feedback from the reviewers.
The main remaining weakness remains in establishing the alignment of LLM-as-judge across various algorithmic recourse tasks; the update that the authors did in Appendix K did not adequately address the weakness (although it is a promising step in the right direction).

---

### Decision · Program_Chairs · 2025-01-22

Reject